# Novel Quadratic Constraints for Extending LipSDP beyond Slope-Restricted Activations

**Patricia Pauli** [*1]     **Aaron Havens** [2]     **Alexandre Araujo** [3]     **Siddharth Garg** [3]
**Farshad Khorrami** [3]     **Frank Allgöwer** [1]     **Bin Hu** [2]

[1] Institute for Systems Theory and Automatic Control, University of Stuttgart
[2] ECE & CSL, University of Illinois Urbana-Champaign
[3] ECE, New York University
[*] Corresponding author. E-Mail: patricia.pauli@ist.uni-stuttgart.de

## Abstract

Recently, semidefinite programming (SDP) techniques have shown great promise in providing accurate Lipschitz bounds for neural networks. Specifically, the LipSDP approach (Fazlyab et al., 2019) has received much attention and provides the least conservative Lipschitz upper bounds that can be computed with polynomial time guarantees. However, one main restriction of LipSDP is that its formulation requires the activation functions to be slope-restricted on $[0, 1]$, preventing its further use for more general activation functions such as GroupSort, MaxMin, and Householder. One can rewrite MaxMin activations for example as residual ReLU networks. However, a direct application of LipSDP to the resultant residual ReLU networks is conservative and even fails in recovering the well-known fact that the MaxMin activation is 1-Lipschitz. Our paper bridges this gap and extends LipSDP beyond slope-restricted activation functions. To this end, we provide novel quadratic constraints for GroupSort, MaxMin, and Householder activations via leveraging their underlying properties such as sum preservation. Our proposed analysis is general and provides a unified approach for estimating $\ell_2$ and $\ell_\infty$ Lipschitz bounds for a rich class of neural network architectures, including non-residual and residual neural networks and implicit models, with GroupSort, MaxMin, and Householder activations. Finally, we illustrate the utility of our approach with a variety of experiments and show that our proposed SDPs generate less conservative Lipschitz bounds in comparison to existing approaches.

## 1 Introduction

For neural network models, the Lipschitz constant is a key sensitivity metric that gives important implications to properties such as robustness, fairness, and generalization (Hein & Andriushchenko, 2017; Tsuzuku et al., 2018; Salman et al., 2019; Leino et al., 2021; Huang et al., 2021; Lee et al., 2020; Bartlett et al., 2017; Miyato et al., 2018; Farnia et al., 2019; Li et al., 2019; Trockman & Kolter, 2021; Singla & Feizi, 2021; Xu et al., 2022; Yu et al., 2022; Meunier et al., 2022; Prach & Lampert, 2022). Since the exact calculation of Lipschitz constants for neural networks is NP-hard in general (Virmaux & Scaman, 2018; Jordan & Dimakis, 2020), relaxation techniques are typically used to compute related upper bounds (Combettes & Pesquet, 2019; Chen et al., 2020; Latorre et al., 2020; Fazlyab et al., 2019; Zhang et al., 2019). A naive Lipschitz bound for a neural network is given by the product of the spectral norm of every layer's weight (Szegedy et al., 2013), yielding quite conservative bounds for multi-layer networks. To provide tighter bounds, Fazlyab et al. (2019) introduced LipSDP, a semidefeinite programming (SDP) based Lipschitz constant estimation technique, that provides state-of-the-art $\ell_2$ Lipschitz bounds in polynomial time. Since its introduction, LipSDP has had a broad impact in many fields, e.g., Lipschitz constant estimation (Hashemi et al., 2021; Shi et al., 2022) and safe neural network controller design (Brunke et al., 2022; Yin et al., 2021). It inspired many follow-up works, extending the framework to more general network architectures (Pauli et al., 2023a) and $\ell_\infty$ bounds (Wang et al., 2022). Beyond Lipschitz constant estimation, many works developed

training methods for Lipschitz neural networks based on LipSDP (Pauli et al., 2021; 2023b; Araujo et al., 2023; Revay et al., 2020; Havens et al., 2023; Wang & Manchester, 2023).

One important assumption of the LipSDP formulation is that it requires the activation functions to be slope-restricted. Recently, new activation functions have been explored that do not have this property. Especially, the MaxMin activation function (Anil et al., 2019) and its generalizations GroupSort and the Householder activation function (Singla et al., 2022) have been popularized for the design of Lipschitz neural networks and have shown great promise as a gradient norm preserving alternative to ReLU, tanh, or sigmoid (Leino et al., 2021; Huang et al., 2021; Hu et al., 2023b; Cohen et al., 2019). MaxMin can equivalently be rewritten as a residual ReLU network, enabling a LipSDP-based analysis based on the slope-restriction property of ReLU. However, as we will discuss in our paper, this description is too conservative to even recover the well-known fact that MaxMin is 1-Lipschitz. This creates an incentive for us to develop tighter quadratic constraints to obtain better Lipschitz bounds beyond slope-restricted activations. Consequently, in this paper, we explore the characteristic properties of MaxMin, GroupSort, and Householder activations to develop *novel quadratic constraints* that are satisfied by these activations.

Based on our new quadratic constraints, our work is the first one to extend LipSDP beyond slope-restricted activations, providing accurate Lipschitz bounds for neural networks with MaxMin, Group-Sort, or Householder activations. More specifically, we use that GroupSort is sum-preserving and our quadratic constraints are also the first ones for sum-preserving elements. Along with our extension of LipSDP to new classes of activation functions, we present a *unified approach* for SDP-based Lipschitz constant estimation that covers a wide range of network architectures and both $\ell_2$ and $\ell_\infty$ bounds. Our work complements the existing SDP-based methods in (Fazlyab et al., 2019; Wang et al., 2022; Araujo et al., 2023; Pauli et al., 2023a; Gramlich et al., 2023b; Revay et al., 2020; Havens et al., 2023; Wang & Manchester, 2023) and further shows the flexibility and versatility of SDP-based approaches, involving the Lipschitz analysis of residual and non-residual networks and implicit learning models with slope-restricted or gradient norm preserving activations.

**Notation.** We denote the set of real $n$-dimensional vectors (with non-negative entries) by $\mathbb{R}^n$ ($\mathbb{R}^n_+$), the set of $m \times n$-dimensional matrices by $\mathbb{R}^{m \times n}$, and the set of $n$-dimensional diagonal matrices with non-negative entries by $\mathbb{D}^n_+$. We denote the $n \times n$ identity matrix by $I_n$ and the $n$-dimensional all ones vector by $\mathbf{1}_n$. Given any vector $x \in \mathbb{R}^n$, we write $\mathrm{diag}(x)$ for a diagonal matrix whose $(i, i)$-th entry is equal to the $i$-th entry of $x$. Clearly, $\mathbf{1}_n^\top x$ is just the sum of all the entries of $x$. We also use $\mathrm{blkdiag}(X_1, X_2, \ldots, X_n)$ to denote a blockdiagonal matrix with blocks $X_1, X_2, \ldots, X_n$. Given two matrices $A$ and $B$, their Kronecker product is denoted by $A \otimes B$. A function $\varphi : \mathbb{R} \to \mathbb{R}$ is slope-restricted on $[\alpha, \beta]$ where $0 \le \alpha < \beta < \infty$ if $\alpha(y - x) \le \varphi(y) - \varphi(x) \le \beta(y - x) \; \forall x, y \in \mathbb{R}$. A mapping $\phi : \mathbb{R}^n \to \mathbb{R}^n$ is slope-restricted on $[\alpha, \beta]$ if each entry of $\phi$ is slope-restricted on $[\alpha, \beta]$.

## 2 PRELIMINARIES

### 2.1 LIPSCHITZ BOUNDS FOR NEURAL NETWORKS: A BRIEF REVIEW

In this section, we briefly review the original LipSDP framework developed in Fazlyab et al. (2019). Suppose we are interested in analyzing the $\ell_2 \to \ell_2$ Lipschitz upper bounds for the following standard $l$-layer feed-forward neural network which maps an input $x$ to the output $f_\theta(x)$:

$$x^0 = x, \quad x^i = \phi(W_i x^{i-1} + b_i) \quad i = 1, \ldots, l-1, \quad f_\theta(x) = W_l x^{l-1} + b_l. \tag{1}$$

Here $W_i \in \mathbb{R}^{n_i \times n_{i-1}}$ and $b_i \in \mathbb{R}^{n_i}$ denote the weight matrix and the bias vector at the $i$-th layer, respectively. Obviously, we have $x \in \mathbb{R}^{n_0}$ and $f_\theta(x) \in \mathbb{R}^{n_l}$ and we collect all characterizing weight and bias terms in $\theta = \{W_i, b_i\}_{i=1}^l$. The goal is to obtain an accurate Lipschitz bound $L > 0$ such that $\|f_\theta(x) - f_\theta(y)\|_2 \le L\|x - y\|_2 \; \forall x, y \in \mathbb{R}^{n_0}$ holds.

**Spectral norm product bound for $\phi$ being 1-Lipschitz.** If the activation function $\phi$ is 1-Lipschitz, then one can use the chain rule to obtain the trivial Lipschitz bound $L = \prod_{i=1}^l \|W_i\|_2$ (Szegedy et al., 2013) . In other words, the product of the spectral norm of every layer's weight provides a simple Lipschitz bound. This spectral norm bound can be efficiently computed using the power iteration method, and the assumption regarding $\phi$ being 1-Lipschitz is quite mild, i. e., almost all activation functions used in practice, e.g., ReLU and MaxMin, satisfy this weak assumption. The downside is

that the spectral norm bound yields conservative results, e.g., ReLU is not only 1-Lipschitz but also slope-restricted on $[0, 1]$, which is exploited in more advanced SDP-based methods.

**LipSDP for $\phi$ being slope-restricted on $[0, 1]$.** If the activation $\phi$ is slope-restricted on $[0, 1]$ in an entry-wise sense, as satisfied, e.g., by ReLU, sigmoid, tanh, one can adopt the quadratic constraint framework from control theory (Megretski & Rantzer, 1997) to formulate LipSDP for an improved Lipschitz analysis (Fazlyab et al., 2019). Given $\{W_i, b_i\}_{i=1}^l$, define $(A, B)$ as

$$
A = \begin{bmatrix} W_1 & 0 & \dots & 0 & 0 \\ 0 & W_2 & \dots & 0 & 0 \\ \vdots & \vdots & \ddots & \vdots & \vdots \\ 0 & 0 & \dots & W_{l-1} & 0 \end{bmatrix}, \ B = \begin{bmatrix} 0 & I_{n_1} & 0 & \dots & 0 \\ 0 & 0 & I_{n_2} & \dots & 0 \\ \vdots & \vdots & \vdots & \ddots & \vdots \\ 0 & 0 & 0 & \dots & I_{n_{l-1}} \end{bmatrix}.
$$

to write the neural network (1) compactly as $Bx = \phi(Ax + b)$ with $x = \begin{bmatrix} x^{0\top} \dots x^{l-1\top} \end{bmatrix}^\top$, $b = \begin{bmatrix} b_1^\top \dots b_l^\top \end{bmatrix}^\top$. Based on (Fazlyab et al., 2019, Theorem 2), if there exists a diagonal positive semidefinite matrix $T \in \mathbb{D}_+^n$, where $n = \sum_{i=1}^l n_i$, such that the following matrix inequality holds[1]

$$
\begin{bmatrix} A \\ B \end{bmatrix}^\top \begin{bmatrix} 0 & T \\ T & -2T \end{bmatrix} \begin{bmatrix} A \\ B \end{bmatrix} + \begin{bmatrix} -\rho I_{n_0} & 0 & \dots & 0 \\ 0 & 0 & \dots & 0 \\ \vdots & \vdots & \ddots & \vdots \\ 0 & 0 & \dots & W_l^\top W_l \end{bmatrix} \preceq 0, \tag{2}
$$

then the neural network (1) with $\phi$ being slope-restricted on $[0, 1]$ is $\sqrt{\rho}$-Lipschitz in the $\ell_2 \to \ell_2$ sense, i.e., $\|f_\theta(x) - f_\theta(y)\|_2 \leq \sqrt{\rho}\|x - y\|_2 \ \forall x, y \in \mathbb{R}^{n_0}$. Minimizing $\rho$ over the decision variables $(\rho, T)$ subject to the matrix inequality constraint (2), i.e.,

$$
\min_{\rho, T \in \mathbb{D}_+^n} \rho \quad \text{s.t.} \quad (2),
$$

is a convex program and exactly the so-called LipSDP, which, in comparison to the spectral norm bound, leads to improved Lipschitz bounds for networks with slope-restricted activations (e.g. ReLU).

**SDPs for $\ell_\infty$ perturbations.** Wang et al. (2022) developed an extension of LipSDP to find Lipschitz bounds for neural networks with slope-restricted activations in the $\ell_\infty \to \ell_1$ sense. Specifically, suppose $f_\theta$ is a scalar output, and the SDPs in Wang et al. (2022) can be used to find $L$ such that $|f_\theta(x) - f_\theta(y)| \leq L\|x - y\|_\infty \quad \forall x, y \in \mathbb{R}^{n_0}$. See Wang et al. (2022) for more details.

## 2.2 GROUPSORT AND HOUSEHOLDER ACTIVATIONS

Recently, there has been a growing interest in using gradient norm preserving activations, particularly MaxMin, to design expressive Lipschitz neural networks (Tanielian & Biau, 2021) that are used, e.g., in the fields of robust neural networks and learning-based control, cmp. Appendix A for details. MaxMin is a special case of the GroupSort activation that we define as follows (Anil et al., 2019). Suppose $\phi : \mathbb{R}^n \to \mathbb{R}^n$ is a GroupSort activation. What $\phi$ does is separating the $n$ preactivations into $N$ groups each of size $n_g$, i.e., $n = Nn_g$, and sorting these groups in ascending order before combining the outputs to a vector of size $n$. GroupSort encompasses two important special cases: MaxMin ($n_g = 2$) and FullSort ($n_g = n$). Another gradient norm preserving activation that also generalizes MaxMin is the Householder activation $\phi : \mathbb{R}^n \to \mathbb{R}^n$ that is applied to subgroups $x \in \mathbb{R}^{n_g}$ (Singla et al., 2022)

$$
\phi(x) = \begin{cases} x & v^\top x > 0 \\ (I_{n_g} - 2vv^\top)x & v^\top x \leq 0 \end{cases}
$$

for all $v : \|v\|_2 = 1$. The parameter $v$ can hence be learned during training and the choice $v = \begin{bmatrix} \frac{\sqrt{2}}{2} & -\frac{\sqrt{2}}{2} \end{bmatrix}^\top$, $n_g = 2$ recovers MaxMin.

Using that the GroupSort and the Householder activation are 1-Lipschitz (Anil et al., 2019; Singla et al., 2022), the spectral norm bound can still be utilized. However, GroupSort and Householder are not slope-restricted on $[0, 1]$. Therefore, LipSDP cannot directly be applied to this case. Our work aims at extending LipSDP and its variants beyond slope-restricted activation functions.

---

[1] The original work in Fazlyab et al. (2019) uses a more general full-block parameterization of $T$. However, as pointed out in Pauli et al. (2021), (Fazlyab et al., 2019, Lemma 1) is actually incorrect, and one has to use diagonal matrices $T$ instead.

## 3 MOTIVATION AND PROBLEM STATEMENT

**Motivating Example.** One may argue that MaxMin activations can be rewritten as residual ReLU networks

$$\text{MaxMin}\left(\begin{bmatrix} x_1 \\ x_2 \end{bmatrix}\right) = \begin{bmatrix} 0 & 1 \\ 0 & 1 \end{bmatrix}\begin{bmatrix} x_1 \\ x_2 \end{bmatrix} + \begin{bmatrix} 1 & 0 \\ 0 & -1 \end{bmatrix}\text{ReLU}\left(\begin{bmatrix} 1 & -1 \\ -1 & 1 \end{bmatrix}\begin{bmatrix} x_1 \\ x_2 \end{bmatrix}\right) = Hx + G\,\text{ReLU}\,(Wx).$$

(3)

Using that ReLU is slope-restricted, one can then formulate a SDP condition

$$\begin{bmatrix} I & 0 \\ H & G \end{bmatrix}^\top \begin{bmatrix} \rho I & 0 \\ 0 & -I \end{bmatrix}\begin{bmatrix} I & 0 \\ H & G \end{bmatrix} \succeq \begin{bmatrix} W & 0 \\ 0 & I \end{bmatrix}^\top \begin{bmatrix} 0 & T \\ T & -2T \end{bmatrix}\begin{bmatrix} W & 0 \\ 0 & I \end{bmatrix},$$

(4)

that implies $\sqrt{\rho}$-Lipschitzness of the MaxMin activation function. The proof of this claim is deferred to Appendix C.1. Minimizing $\rho$ over $(\rho, T)$ subject to matrix inequality (4) yields $\rho = 2$. This shows that applying LipSDP to the residual ReLU network equivalent to MaxMin is too conservative to recover the well-known fact that MaxMin is 1-Lipschitz. To get improved Lipschitz bounds, we necessitate more sophisticated qudratic constraints for MaxMin and its generalizations GroupSort and Householder, which we derive in the next section.

**Problem statement.** In this paper, we are concerned with *Lipschitz constant estimation* for neural networks with *MaxMin*, *GroupSort*, and *Householder* activations. We give a brief preview of our approach here. Similar to LipSDP, we also follow the quadratic constraint approach from control theory (Megretski & Rantzer, 1997) to obtain a matrix inequality that implies Lipschitz continuity for a neural network. We provide more details on the quadratic constraint framework in Appendix B. Notice that the matrix inequality (2) used in LipSDP is a special case of

$$\begin{bmatrix} A \\ B \end{bmatrix}^\top X \begin{bmatrix} A \\ B \end{bmatrix} + \begin{bmatrix} -\rho I_{n_0} & 0 & \dots & 0 \\ 0 & 0 & \dots & 0 \\ \vdots & \vdots & \ddots & \vdots \\ 0 & 0 & \dots & W_l^\top W_l \end{bmatrix} \preceq 0,$$

(5)

with $X = \begin{bmatrix} 0 & T \\ T & -2T \end{bmatrix}$ for $[0,1]$-slope-restricted activations.

Developing a proper choice of $X$ beyond slope-restricted activations is the goal of this paper. In doing so, we explore the input-output properties of GroupSort and Householder activations to set up $X$ such that the matrix inequality (5) can be modified to give state-of-the-art Lipschitz bounds for neural networks with important and popular activations that are not slope-restricted.

## 4 MAIN RESULTS: SDPs FOR GROUPSORT AND HOUSEHOLDER ACTIVATIONS

In this section, we develop novel SDPs for analyzing Lipschitz bounds of neural networks with MaxMin, GroupSort, and Householder activations. Our developments are based on non-trivial constructions of quadratic constraints capturing the input-output properties of the respective activation function and control-theoretic arguments (Megretski & Rantzer, 1997).

### 4.1 QUADRATIC CONSTRAINTS FOR GROUPSORT AND HOUSEHOLDER ACTIVATIONS

First, we present quadratic constraints characterizing the key input-output properties, i.e., 1-Lipschitzness and sum preservation, of the GroupSort function. Our results are summarized below.

**Lemma 1.** *Consider a GroupSort activation $\phi : \mathbb{R}^n \to \mathbb{R}^n$ with group size $n_g$. Let $N = \frac{n}{n_g}$. Given any $\lambda \in \mathbb{R}_+^N$ and $\gamma, \nu, \tau \in \mathbb{R}^N$, the following inequality holds for all $x, y \in \mathbb{R}^n$:*

$$\begin{bmatrix} x - y \\ \phi(x) - \phi(y) \end{bmatrix}^\top \begin{bmatrix} T - 2S & P + S \\ P + S & -T - 2P \end{bmatrix}\begin{bmatrix} x - y \\ \phi(x) - \phi(y) \end{bmatrix} \geq 0,$$

(6)

*where $(T, S, P)$ are given as*

$$T := \text{diag}(\lambda) \otimes I_{n_g} + \text{diag}(\gamma) \otimes (\mathbf{1}_{n_g}\mathbf{1}_{n_g}^\top),$$
$$P := \text{diag}(\nu) \otimes (\mathbf{1}_{n_g}\mathbf{1}_{n_g}^\top),$$
$$S := \text{diag}(\tau) \otimes (\mathbf{1}_{n_g}\mathbf{1}_{n_g}^\top).$$

(7)

All the detailed technical proofs are deferred to Appendix C. To illustrate the technical essence, we briefly sketch a proof outline here.

**Proof sketch.** Let $e_i$ denote a $N$-dimensional unit vector whose $i$-th entry is 1 and all other entries are 0. Since GroupSort preserves the sum within each subgroup, $(e_i^\top \otimes \mathbf{1}_{n_g}^\top)x = (e_i^\top \otimes \mathbf{1}_{n_g}^\top)\phi(x)$ holds for all $i = 1, \dots, N$. This leads to the following key equality:

$$(x-y)^\top ((e_i e_i^\top) \otimes (\mathbf{1}_{n_g}\mathbf{1}_{n_g}^\top))(x-y) = (\phi(x)-\phi(y))^\top((e_i e_i^\top) \otimes (\mathbf{1}_{n_g}\mathbf{1}_{n_g}^\top))(\phi(x)-\phi(y))$$

$$= (\phi(x)-\phi(y))^\top((e_i e_i^\top) \otimes (\mathbf{1}_{n_g}\mathbf{1}_{n_g}^\top))(x-y) = (x-y)^\top((e_i e_i^\top) \otimes (\mathbf{1}_{n_g}\mathbf{1}_{n_g}^\top))(\phi(x)-\phi(y)),$$

which can be used to verify that (6) is equivalent to

$$\begin{bmatrix} x-y \\ \phi(x)-\phi(y) \end{bmatrix}^\top \begin{bmatrix} \text{diag}(\lambda) \otimes I_{n_g} & 0 \\ 0 & -\text{diag}(\lambda) \otimes I_{n_g} \end{bmatrix} \begin{bmatrix} x-y \\ \phi(x)-\phi(y) \end{bmatrix} \geq 0.$$

Finally, the above inequality holds because the sorting behavior in each subgroup is 1-Lipschitz, i.e., $\|(e_i^\top \otimes I_{n_g})(x-y)\|_2^2 \geq \|(e_i^\top \otimes I_{n_g})(\phi(x)-\phi(y))\|_2^2$ for all $i = 1, \dots, N$. $\qquad\square$

Based on the above proof outline, we can see that the construction of the quadratic constraint (6) relies on the key properties that the sorting in every subgroup is 1-Lipschitz and sum-preserving. This is significantly different from the original sector-bounded quadratic constraint for slope-restricted nonlinearities in Fazlyab et al. (2019).

In the following, we present quadratic constraints for the group-wise applied Householder activation.

**Lemma 2.** *Consider a Householder activation* $\phi : \mathbb{R}^n \to \mathbb{R}^n$ *with group size* $n_g$. *Let* $N = \frac{n}{n_g}$. *Given any* $\lambda \in \mathbb{R}_+^N$ *and* $\gamma, \nu, \tau \in \mathbb{R}^N$, *the inequality (6) holds for all* $x, y \in \mathbb{R}^n$, *where* $(T, S, P)$ *are given as*

$$\begin{aligned} T &:= \text{diag}(\lambda) \otimes I_{n_g} + \text{diag}(\gamma) \otimes (I_{n_g} - vv^\top), \\ P &:= \text{diag}(\nu) \otimes (I_{n_g} - vv^\top), \\ S &:= \text{diag}(\tau) \otimes (I_{n_g} - vv^\top). \end{aligned} \qquad (8)$$

Using the novel quadratic constraint (6) with $(T, S, P)$ either defined as in (7) or in (8), we can formulate a SDP subject to (5) with $X$ being set up as $X = \begin{bmatrix} T-2S & P+S \\ P+S & -T-2P \end{bmatrix}$ to estimate the Lipschitz constant of a GroupSort/Householder neural network. We will discuss such results next.

## 4.2 $\ell_2 \to \ell_2$ LIPSCHITZ BOUNDS FOR GROUPSORT/HOUSEHOLDER NEURAL NETWORKS

Based on Lemmas 1 and 2, we develop a new SDP condition to analyze neural networks with GroupSort and Householder activations. To this end, we establish the following result.

**Theorem 1.** *Consider the* $l$*-layer fully-connected neural network (1), where* $\phi$ *is GroupSort (Householder) with the same group size* $n_g$ *in all layers. Let* $N = \frac{\sum_{i=1}^{l-1} n_i}{n_g}$ *be the total number of groups. If there exist* $\rho > 0$, $\lambda \in \mathbb{R}_+^N$, *and* $\gamma, \nu, \tau \in \mathbb{R}^N$ *such that the following matrix inequality holds*

$$M(\rho, P, S, T) := \begin{bmatrix} A \\ B \end{bmatrix}^\top \begin{bmatrix} T-2S & P+S \\ P+S & -T-2P \end{bmatrix} \begin{bmatrix} A \\ B \end{bmatrix} + \begin{bmatrix} -\rho I_{n_0} & 0 & \dots & 0 \\ 0 & 0 & \dots & 0 \\ \vdots & \vdots & \ddots & \vdots \\ 0 & 0 & \dots & W_l^\top W_l \end{bmatrix} \preceq 0, \quad (9)$$

*where* $(T, S, P)$ *are defined by (7) (eq. (8)), then (1) is* $\sqrt{\rho}$*-Lipschitz in the* $\ell_2 \to \ell_2$ *sense.*

The proof of the above result is based on standard quadratic constraint arguments and is included in Appendix C.4. Based on this result, the solution $\sqrt{\rho}$ of the SDP

$$\min_{\rho, T, S, P} \rho \quad \text{s.t.} \quad M(\rho, T, S, P) \preceq 0, \qquad (10)$$

gives an accurate Lipschitz bound for neural networks with GroupSort or Householder activations. In our practical implementation, we set $S = P = 0$ for improved scalability, i.e., to reduce the number of decision variables and to exploit the structure of the constraint (9). See Appendix D.1 for details. Empirical tests indicate that the choice $S = P = 0$ yields the same results as without this consraint. In what follows, we discuss several important aspects of SDP-based Lipschitz constant estimation for GroupSort and Householder activations.

**Simplifications for the single-layer neural network.** If the neural network is single-layer, i.e., $f_\theta(x) = W_2\phi(W_1 x + b_1) + b_2$ with $\phi$ being GroupSort/Householder with group size $n_g$, we can simplify our SDP condition (9) as

$$\begin{bmatrix} W_1^\top T W_1 - \rho I & 0 \\ 0 & -T + W_2^\top W_2 \end{bmatrix} \preceq 0,$$

setting $S = P = 0$ as discussed before. One can compare the above result to its LipSDP counterpart (Fazlyab et al., 2019, Theorem 1) and realize that the only main difference is that Lemma 1 is used to replace (Fazlyab et al., 2019, Lemma 1) for setting up $X$.

**MaxMin activations.** In practice, MaxMin is typically used where $n_g = 2$. In this case, the number of the decision variables used to parameterize $T$ is exactly $\sum_{i=1}^{l-1} n_i$. This is consistent with LipSDP, which uses the same number of decision variables to address slope-restricted activations.

**GroupSort and norm-constrained weights.** GroupSort activations are oftentimes used in combination with gradient norm preserving layers. For this special case the Lipschitz constant 1 is tight and hence there is no need for Lipschitz constant estimation. However, many MaxMin neural networks are trained without weight constraints as they have shown to be prohibitive in training (Leino et al., 2021; Hu et al., 2023b; Huang et al., 2021; Cohen et al., 2019). For such neural networks our approach provides more accurate Lipschitz bounds than existing approaches.

**Adaptions to convolutional neural networks.** Theorem 1 can be extended to convolutional neural networks using ideas from Pauli et al. (2023a); Gramlich et al. (2023b). The key idea in (Pauli et al., 2023a; Gramlich et al., 2023b) is the formulation of convolutional layers via a state space representation, which enables the derivation of non-sparse and scalable SDPs for Lipschitz constant estimation for convolutional neural networks. We briefly streamline the method in Appendix F.2.

### 4.3 $\ell_\infty \to \ell_1$ LIPSCHITZ BOUNDS FOR GROUPSORT/HOUSEHOLDER NEURAL NETWORKS

In this section, we will combine Lemmas 1 and 2 with the results in Wang et al. (2022) to develop SDPs for the $\ell_\infty \to \ell_1$ analysis of neural networks with GroupSort/Householder activations. Similar to Wang et al. (2022), we assume that $f_\theta(x)$ has a scalar output. Suppose $\phi$ is a GroupSort/Householder activation with group size $n_g$. We are interested in calculating $L$ such that $|f_\theta(x) - f_\theta(y)| \leq L\|x - y\|_\infty \ \forall x, y \in \mathbb{R}^{n_0}$. It is well known that the $\ell_2 \to \ell_2$ Lipschitz bounds can be transferred into $\ell_\infty \to \ell_1$ Lipschitz bounds via some standard scaling argument based on the equivalence of norms. However, such a naive approach is too conservative in practice (Latorre et al., 2020). In this section, we develop novel SDPs for more accurate calculations of $\ell_\infty \to \ell_1$ Lipschitz bounds.

**Single-layer case.** For simplicity, we first discuss the single-layer network case, i.e., $f_\theta(x) = W_2\phi(W_1 x + b_1) + b_2$ with $\phi$ being GroupSort/Householder with group size $n_g$. Since the output $f_\theta(x)$ is assumed to be a scalar, we have $W_2 \in \mathbb{R}^{1 \times n_1}$. Obviously, the tightest Lipschitz bound is given by

$$L_{\min} = \max_{x,y} \frac{|f_\theta(x) - f_\theta(y)|}{\|x - y\|_\infty}. \tag{11}$$

Then for any $L \geq L_{\min}$, we will naturally have $|f_\theta(x) - f_\theta(y)| \leq L\|x - y\|_\infty \ \forall x, y$. Based on Lemma 1, one upper bound for $L_{\min}$ is provided by the solution of the following problem:

$$\max_{\Delta x, \Delta v} \frac{|W_2 \Delta v|}{\|\Delta x\|_\infty} \quad \text{s.t.} \quad \begin{bmatrix} W_1 \Delta x \\ \Delta v \end{bmatrix}^\top \begin{bmatrix} T - 2S & P + S \\ P + S & -T - 2P \end{bmatrix} \begin{bmatrix} W_1 \Delta x \\ \Delta v \end{bmatrix} \geq 0, \tag{12}$$

where $\Delta v = \phi(W_1 x + b_1) - \phi(W_1 y + b_1)$, $\Delta x = x - y$, and $(T, S, P)$ are given by (7) with some $\gamma, \nu, \tau \in \mathbb{R}^N$, $\lambda \in \mathbb{R}_+^N$. A similar argument has been used in (Wang et al., 2022, Section 5). The difference here is that the constraint in the optimization problem (12) is given by our new quadratic constraint specialized for GroupSort/Householder activations (Lemma 1/2). We note that if we scale $\Delta x$ and $\Delta v$ with some common factor, the constraint in problem (12) is maintained due to its quadratic form, and the objective ratio remains unchanged. Hence the problem (12) is scaling-invariant, making it equivalent to the following problem (the absolute value in the objective is removed since scaling $\Delta v$ with $-1$ does not affect feasibility)

$$\max_{\Delta x, \Delta v} W_2 \Delta v \quad \text{s.t.} \quad \begin{bmatrix} W_1 \Delta x \\ \Delta v \end{bmatrix}^\top \begin{bmatrix} T - 2S & P + S \\ P + S & -T - 2P \end{bmatrix} \begin{bmatrix} W_1 \Delta x \\ \Delta v \end{bmatrix} \geq 0, \ \|\Delta x\|_\infty = 1.$$

A simple upper bound for the above problem is provided by replacing the equality constraint with an inequality $\|\Delta x\|_\infty \leq 1$. Therefore, we know $L_{\min}$ can be upper bounded by the solution of the following problem:

$$\max_{\Delta x, \Delta v} W_2 \Delta v \quad \text{s.t.} \quad \begin{bmatrix} \Delta x \\ \Delta v \end{bmatrix}^\top \begin{bmatrix} W_1^\top (T - 2S) W_1 & W_1^\top (P + S) \\ (P + S) W_1 & -T - 2P \end{bmatrix} \begin{bmatrix} \Delta x \\ \Delta v \end{bmatrix} \geq 0, \ \|\Delta x\|_\infty \leq 1. \quad (13)$$

Finally, we note that $\|\Delta x\|_\infty \leq 1$ can be equivalently rewritten as the following quadratic constraints $(e_j^\top \Delta x)^2 \leq 1$ for $j = 1, \ldots, n_0$ (Wang et al., 2022). Now we introduce the dual variable $\mu \in \mathbb{R}_+^{n_0}$ and obtain the following SDP condition (which can be viewed as the dual program of (13)).

**Theorem 2.** *Consider a single-layer neural network described by* $f_\theta(x) = W_2 \phi(W_1 x + b_1) + b_2$, *where* $\phi : \mathbb{R}^{n_1} \to \mathbb{R}^{n_1}$ *is GroupSort (Householder) with group size* $n_g$. *Denote* $N = \frac{n_1}{n_g}$. *Suppose there exist* $\rho > 0$, $\gamma, \nu, \tau \in \mathbb{R}^N$, $\lambda \in \mathbb{R}_+^N$, *and* $\mu \in \mathbb{R}_+^{n_0}$ *such that*

$$\begin{bmatrix} \mathbf{1}_{n_0}^\top \mu - 2\rho & 0 & W_2 \\ 0 & W_1^\top (T - 2S) W_1 - Q & W_1^\top (P + S) \\ W_2^\top & (P + S) W_1 & -T - 2P \end{bmatrix} \preceq 0, \quad (14)$$

*where* $(T, S, P)$ *are given by (7) (eq. (8)), and* $Q := \mathrm{diag}(\mu)$. *Then* $|f_\theta(x) - f_\theta(y)| \leq \rho \|x - y\|_\infty \forall x, y \in \mathbb{R}^{n_0}$.

**Multi-layer case.** The above analysis can easily be extended to address the multi-layer network case with GroupSort/Householder activations. Again, we adopt the setting in Wang et al. (2022) and choose $n_l = 1$. By slightly modifying the above analysis, we then obtain the following result.

**Theorem 3.** *Consider the* $\ell$-*layer feedforward neural network (1), where* $\phi$ *is GroupSort (Householder) with group size* $n_g$ *in all layers. Let* $N = \frac{\sum_{i=1}^{l-1} n_i}{n_g}$ *be the total number of groups. Suppose there exist* $\rho > 0$, $\gamma, \nu, \tau \in \mathbb{R}^N$, $\lambda \in \mathbb{R}_+^N$, *and* $\mu \in \mathbb{R}_+^{n_0}$ *such that*

$$M(\rho, T) = \begin{bmatrix} 0 & A \\ 0 & B \end{bmatrix}^\top \begin{bmatrix} T - 2S & P + S \\ P + S & -T - 2P \end{bmatrix} \begin{bmatrix} 0 & A \\ 0 & B \end{bmatrix} + \begin{bmatrix} \mathbf{1}_{n_0}^\top \mu - 2\rho & 0 & \ldots & W_l \\ 0 & -Q & \ldots & 0 \\ \vdots & \vdots & \ddots & \vdots \\ W_l^\top & 0 & \ldots & 0 \end{bmatrix} \preceq 0,$$

$$(15)$$

*where* $(T, S, P)$ *are given by (7) (eq. (8)), and* $Q := \mathrm{diag}(\mu)$. *Then* $|f_\theta(x) - f_\theta(y)| \leq \rho \|x - y\|_\infty \forall x, y \in \mathbb{R}^{n_0}$.

Minimizing $\rho$ subject to the constraint (15) yields a SDP problem. For large-scale implementations, we again set $S = P = 0$ and implement the SDP as specified in Appendix D.2 to solve it more efficiently. Based on preliminary testing, this implementation does not introduce conservatism over the original SDP formulation with non-zero $(S, P)$.

## 4.4 FURTHER GENERALIZATIONS: RESIDUAL NETWORKS AND IMPLICIT MODELS

In this section, we briefly discuss how our new quadratic constraints in Lemmas 1 and 2 can be used to generalize various existing SDPs to handle residual networks and implicit learning models with MaxMin, GroupSort, and Householder activations. Due to space limitations, we keep our discussion brief. More details can be found in Appendix F.

**Residual neural networks.** We consider a residual neural network

$$x^0 = x, \quad x^i = x^{i-1} + G_i \phi(W_i x^{i-1} + b_i) \quad i = 1, \ldots, l, \quad f_\theta(x) = x^l, \quad (16)$$

referring the reader to Appendix F.1 for the single-layer case. Suppose $\phi$ is GroupSort/Householder with the same group size $n_g$ for all layers. For simplicity, we define the matrices $\tilde{A}$ and $\tilde{B}$ as

$$\widetilde{A} = \begin{bmatrix} W_1 & 0 & \ldots & 0 & 0 \\ W_2 & W_2 G_1 & \ldots & 0 & 0 \\ \vdots & \vdots & \ddots & \vdots & \vdots \\ W_l & W_l G_1 & \ldots & W_l G_{l-1} & 0 \end{bmatrix}, \quad \widetilde{B} = \begin{bmatrix} 0 & I_{n_1} & 0 & \ldots & 0 \\ 0 & 0 & I_{n_2} & \ldots & 0 \\ \vdots & \vdots & \vdots & \ddots & \vdots \\ 0 & 0 & 0 & \ldots & I_{n_l} \end{bmatrix}$$

Table 1: This table presents the results of our LipSDP-NSR method on several feed-forward neural networks with MaxMin activations trained on MNIST with more than 96% accuracy, using $\ell_2$ adversarial training. We compare against several metrics which are described in detail in Section 5. The computation time in seconds is shown in parentheses and included in Appendix D.3 for the naive bounds. Similar to Wang et al. (2022), we report the $\ell_\infty$ bounds with respect to label 8 to consider scalar outputs.

| Model (units per layer) | | $\ell_2$ Sample | FGL | LipSDP-NSR | LipSDP-RR | MP | $\ell_\infty$ Sample | FGL | LipSDP-NSR | LipSDP-RR | Norm Eq. | MP |
|---|---|---|---|---|---|---|---|---|---|---|---|---|
| 2-layer | 16 units | 20.98 | 22.02 | 22.59 (54) | 23.92 (18) | 23.46 | 142.42 | 153.22 | 202.82 (94) | 207.3 (34) | 632.52 | 543 |
| | 32 units | 34.64 | 37.75 | 38.55 (86) | 40.28 (24) | 42.46 | 307.95 | 318.91 | 368.68 (128) | 404.8 (44) | 1079.4 | 1140 |
| | 64 units | 63.35 | - | 73.77 (160) | 74.31 (23) | 83.98 | 653.07 | - | 885.89 (147) | 870.8 (38) | 2065.6 | 3572 |
| | 128 units | 55.44 | - | 70.87 (182) | 72.18 (55) | 80.02 | 667.55 | - | 1058.8 (236) | 1033.1 (77) | 1984.4 | 4889 |
| 5-layer | 32 units | 106.35 | - | 224.53 (53) | 388.0 (69) | 298.41 | 932.89 | - | 2969.5 (96) | 4283.1 (137) | 6286.8 | 522712 |
| | 64 units | 232.21 | - | 386.81 (360) | 623.4 (228) | 520.73 | 2187.32 | - | 6379.8 (190) | 8321.3 (416) | 10831 | 1337103 |
| 8-layer | 32 units | 113.34 | - | 401.03 (58) | 1078.5 (178) | 780 | 873.65 | - | 4303.9 (120) | | 11229 | 2.01E+07 |
| | 64 units | 257.63 | - | 722.34 (116) | 1683.7 (1426) | 1241 | 2147.64 | - | 9632.2 (188) | 1.886E+04 (2495) | 20226 | 1.22E+08 |
| | 128 units | 258.85 | - | 714.14 (219) | - | 1167 | 2496.75 | - | 10164 (318) | - | 19996 | 3.35E+08 |
| | 256 units | 207.08 | - | 642.07 (612) | - | 934 | 1845.13 | - | 9256.3 (769) | - | 17978 | 1.13E+09 |
| 18-layer | 32 units | 267.32 | - | 17432 (64) | - | 73405 | 1693.49 | - | 1.7599E+05 (78) | - | 4.881E+05 | 2.23E+13 |
| | 64 units | 523.85 | - | 22842 (129) | - | 84384 | 2037.21 | - | 1.9367E+05 (150) | - | 6.3958E+05 | 1.00E+15 |
| | 128 units | 536.04 | - | 24271 (300) | - | 74167 | 685.88 | - | 1.9689E+05 (331) | - | 6.7959E+05 | 2.79E+16 |

to compactly describe the model (16) as $\widetilde{B}x = \phi(\widetilde{A}x + b)$ with $b = \begin{bmatrix} b_1^\top \dots b_l^\top \end{bmatrix}^\top$. Then we obtain the following theorem.

**Theorem 4.** *Consider the multi-layer residual network (16), where $\phi$ is GroupSort (Householder) with the same group size $n_g$ for all layers. Define $N = \frac{\sum_{i=1}^l n_i}{n_g}$. Suppose there exist $\rho > 0$, $\gamma, \nu, \tau \in \mathbb{R}^N$ and $\lambda \in \mathbb{R}_+^N$ such that*

$$\begin{bmatrix} \widetilde{A} \\ \widetilde{B} \end{bmatrix}^\top \begin{bmatrix} T - 2S & P + S \\ P + S & -T - 2P \end{bmatrix} \begin{bmatrix} \widetilde{A} \\ \widetilde{B} \end{bmatrix} + \begin{bmatrix} (1 - \rho)I_{n_0} & G_1 & \dots & G_l \\ G_1^\top & G_1^\top G_1 & \dots & G_1^\top G_l \\ \vdots & \vdots & \ddots & \vdots \\ G_l^\top & G_l^\top G_1 & \dots & G_l^\top G_l \end{bmatrix} \preceq 0, \quad (17)$$

*where $(T, S, P)$ are given by (7) (eq. (8)). Then (16) is $\sqrt{\rho}$-Lipschitz in the $\ell_2 \to \ell_2$ sense.*

From empirical studies we know that, when implementing the SDP, we can either set $S = 0$ or $P = 0$ (but not both), yielding the same SDP solution as with non-zero $(S, P)$.

**Implicit learning models.** With our proposed quadratic constraints, we can also formulate SDPs for the Lipschitz analysis of implicit models with GroupSort/Householder activations. When the activation function is slope-restricted on $[0, 1]$, SDPs can be formulated for the Lipschitz analysis of both deep equilibrium models (DEQs) (Bai et al., 2019) and neural ordinary differential equations (Neural ODEs) (Chen et al., 2018). For example, a variant of LipSDP has been formulated in (Revay et al., 2020, Theorem 2) to address the Lipschitz bounds of DEQs with $[0, 1]$-slope-restricted activations based on the quadratic constraint in (Fazlyab et al., 2019, Lemma 1). When GroupSort/Householder is used, we can easily modify the result in (Revay et al., 2020, Theorem 2) to obtain a new SPD for DEQs with GroupSort/Householder activations via replacing the slope-restricted quadratic constraints with (6) in Lemma 1/2. Similar results can also be obtained for neural ODEs. Due to the space limit, we defer the detailed discussion of such extensions to Appendices F.3 and F.4.

## 5 NUMERICAL EXPERIMENTS

In this section, we present numerical results demonstrating the effectiveness of our proposed SDP, whose solution we denote by LipSDP-NSR. Additional numerical results including experiments on residual neural networks are included in Appendix E. Code to reproduce all experiments is provided at `https://github.com/ppauli/QCs_MaxMin`. Following the same experimental setting as Wang et al. (2022), we evaluate our approach on several MaxMin neural networks trained on the MNIST dataset (LeCun & Cortes, 2010) and compare LipSDP-NSR to four other measurements:

- **Sample**: The sampling approach computes the norm of the gradient (or the Jacobian) on 200k random samples from the input space. This measurement is easy to compute and provides a lower bound on the true Lipschitz constant.

- **Formal global Lipschitz (FGL)**: This metric is computed by iterating over all (including infeasible) activation patterns. Despite being an upper bound on the true Lipschitz constant, this value is accurate. However, it can only be computed for small networks as it is an exponential-time search. We provide computation details for FGL on GroupSort networks below.
- **LipSDP-RR** We formulate MaxMin activations as residual ReLU networks (3) and solve an SDP using slope-restriction quadratic constraints.
- **Equivalence of Norms (Norm Eq.)**: This bound on the $\ell_\infty$-Lipschitz bound is obtained using the equivalence of norms, i. e., $\sqrt{n_0} \cdot \ell_2$-LipSDP-NSR.
- **Matrix-Product (MP)**: Naive upper bound of the product of the spectral norms of the weights.

**Implementation details of FGL.** As shown in Virmaux & Scaman (2018), based on Rademacher's theorem, if $f$ is Lipschitz continuous, then $f$ is almost everywhere differentiable, and $L_f = \sup_x \|\nabla f(x)\|_q$. From this result and based on the chain rule, we can write the Lipschitz constant of the network $f$ as

$$L_f = \sup_x \|W_l \nabla\phi(f_{l-1}(x))W_{l-1}\cdots\nabla\phi(f_1(x))W_1\|_q, \tag{18}$$

where $\phi$ is the GroupSort activation with group size $n_g$ and $f_i(x) = W_i x^{i-1} + b_i$, $i = 1,\ldots, l-1$. Given that GroupSort is simply a rearrangement of the values of the activation, the Jacobian $\nabla\phi(f(\cdot))$ is a permutation matrix. Let $\mathcal{P}_{n_g}$ be the set of permutation matrices of size $n_g \times n_g$, $N = \frac{n}{n_g}$ and let us define the set of GroupSort permutations:

$$\widetilde{\mathcal{P}}_{n_g} = \left\{ \text{blkdiag}(P_1,\ldots,P_N) \mid P_1\ldots,P_N \in \mathcal{P}_{n_g} \right\}. \tag{19}$$

Following the same reasoning as Wang et al. (2022) and Virmaux & Scaman (2018), we can upper bound $L_f$ as follows:

$$L_f \leq \max_{\widetilde{P}_1,\ldots,\widetilde{P}_{l-1} \in \widetilde{\mathcal{P}}_{n_g}} \left\| W_l \widetilde{P}_{l-1} W_{l-1}\cdots\widetilde{P}_1 W_1 \right\|_q \quad := \text{FGL}. \tag{20}$$

**Implementation details of LipSDP-NSR & discussion of the results.** We solve problem (10) and an SDP that minimizes $\rho$ subject to (15), using YALMIP (Löfberg, 2004) with the solver Mosek (MOSEK ApS, 2020) in Matlab on a standard i7 note book. Our results and the other bounds are summarized in Table 1. For all networks, LipSDP-NSR clearly outperforms MP and the equivalence of norms bound. More detailed discussions are given in Appendices D and E.

**Limitations.** SDP-based Lipschitz bounds are the best bounds that can be computed in polynomial time. One limitation is their scalability to large neural networks that has been tackled by structure exploiting approaches (Newton & Papachristodoulou, 2023; Xue et al., 2022; Pauli et al., 2023a). Our implementation incorporates such approaches and we provide implementation details in Appendix D. In addition, we see potential that new SDP solvers or tailored SDP solvers (Parrilo, 2000; Gramlich et al., 2023a) will mitigate this limitation in the future. Finally, the concurrent work in Wang et al. (2024) has developed the exact-penalty form of LipSDP for the slope-restricted activation case, enabling the use of first-order optimization methods for scaling up the computation. We believe that the argument in Wang et al. (2024) can be modified to improve the scalability of our proposed SDP conditions.

## 6 CONCLUSION

We proposed a SDP-based method for computing Lipschitz bounds of neural networks with MaxMin, GroupSort and Householder activations with respect to $\ell_2$- and $\ell_\infty$-perturbations. To this end, we derived novel quadratic constraints for GroupSort and Householder activations and set up SDP conditions that generalize LipSDP and its variants beyond the original slope-restricted activation setting. Our analysis method covers a large family of network architectures, including residual, non-residual, and implicit models, bridging the gap between SDP-based Lipschitz analysis and this important new class of activation functions. Future research includes finding specific quadratic constraints for other general activation functions and nonlinear parts in the neural network.

## ACKNOWLEDGEMENTS

P. Pauli and F. Allgöwer are funded by Deutsche Forschungsgemeinschaft (DFG, German Research Foundation) under Germany's Excellence Strategy - EXC 2075 - 390740016 and under grant 468094890. A. Havens and B. Hu are generously supported by the NSF award CAREER-2048168, the AFOSR award FA9550-23-1-0732, and the IBM/IIDAI award 110662-01. A. Araujo, S. Garg, and F. Khorrami are supported in part by the Army Research Office under grant number W911NF-21-1-0155 and by the New York University Abu Dhabi (NYUAD) Center for Artificial Intelligence and Robotics, funded by Tamkeen under the NYUAD Research Institute Award CG010. P. Pauli also acknowledges the support by the Stuttgart Center for Simulation Science (SimTech) and the support by the International Max Planck Research School for Intelligent Systems (IMPRS-IS).

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

## A    APPLICATIONS OF MAXMIN NEURAL NETWORKS

GroupSort and Householder activations have been introduced as gradient-norm preserving activation functions. This convenient property lead to the use of neural networks with GroupSort and Householder activations to fit Lipschitz functions (Anil et al., 2019; Tanielian & Biau, 2021; Cohen et al., 2019) and to design robust neural networks (Singla et al., 2022; Huang et al., 2021). Using MaxMin neural networks, Huang et al. (2021); Leino et al. (2021); Hu et al. (2023b;a); Losch et al. (2023) provide state-of-the-art verified-robust accuracy, that they compute based on information of the Lipschitz constant and the prediction margin of the trained networks.

Another application of Lipschitz neural networks is in learning-based control with safety, robustness and stability guarantees (Aswani et al., 2013; Berkenkamp et al., 2017; Jin & Lavaei, 2020; Shi et al., 2019; Fabiani & Goulart, 2022; Yion & Wu, 2023), for example using MaxMin activations (Yion & Wu, 2023). LipSDP for MaxMin can potentially reduce the conservatism in the stability and robustness analysis of such MaxMin network controllers.

## B    THE QUADRATIC CONSTRAINT FRAMEWORK

The quadratic constraint framework stems from the control community and was developed starting in the 1960s (Megretski & Rantzer, 1997). In the control literature, quadratic constraints are used to abstract elements of a dynamical system that cause trouble in the analysis, e.g., nonlinearities, uncertainties, or time delays. This abstraction then enables the analysis of systems which include such troublesome elements. The quadratic constraint framework is generally applicable, as long as the troublesome element can be abstracted by quadratic constraints. For example, Kao & Rantzer (2007) introduce quadratic constraints for a so-called "delay-difference" operator to describe varying time delays, Pfifer & Seiler (2015) use a geometric interpretation to derive quadratic constraints for delayed nonlinear and parameter-varying systems, and Carrasco et al. (2016) summarizes the development of Zames-Falb multipliers for slope-restricted nonlinearites based on frequency domain arguments. All previous papers on LipSDP borrow existing quadratic constraints for slope-restricted nonlinearities from the control literature that are not satisfied by MaxMin, GroupSort, and Householder activations. This being said, our paper is the first that successfully derives quadratic constraints for GroupSort and Householder activations, and further, we are the first to formulate quadratic constraints for sum-preserving elements. The idea we used to derive these quadratic constraints is creative in the sense that it is very different from all the existing quadratic constraint derivations in the large body of control literature. The difficulty in deriving new quadratic constraints lies in identifying properties and characteristics of the troublesome element, in our case a multivariate nonlinearity GroupSort or Householder, that can (i) be formulated in a quadratic form and (ii) whose quadratic constraint formulation is tight and descriptive enough to lead to improvements in the analysis. Please see Appendices C.3 and C.3 for the technical details of our derivation. We use our novel quadratic constraints to solve the problem of Lipschitz constant estimation. However, the modularity of the quadratic constraint framework allows to, as well, address different problems, e.g., safety verification of a MaxMin neural network similar to (Fazlyab et al., 2020; Newton & Papachristodoulou, 2021) or the stability of a feedback loop that includes a MaxMin neural network, as done in (Yin et al., 2021) for slope-restricted activations.

## C    TECHNICAL PROOFS OF MAIN RESULTS

### C.1    PROOF FOR MOTIVATING EXAMPLE

In Section 3, we claim that the matrix inequality (4) implies $\sqrt{\rho}$-Lipschitzness for the residual ReLU network that we obtained from rewriting MaxMin. In the following, we prove this claim. We left and right multiply (4) with $\begin{bmatrix} (x-y)^\top & (\mathrm{ReLU}(Wx) - \mathrm{ReLU}(Wy))^\top \end{bmatrix}$ and its transpose, respectively, and obtain

$$\begin{bmatrix} x-y \\ \mathrm{ReLU}(Wx) - \mathrm{ReLU}(Wy) \end{bmatrix}^\top \begin{bmatrix} I & 0 \\ H & G \end{bmatrix}^\top \begin{bmatrix} \rho I & 0 \\ 0 & -I \end{bmatrix} \begin{bmatrix} I & 0 \\ H & G \end{bmatrix} \begin{bmatrix} x-y \\ \mathrm{ReLU}(Wx) - \mathrm{ReLU}(Wy) \end{bmatrix}$$
$$\geq \begin{bmatrix} x-y \\ \mathrm{ReLU}(Wx) - \mathrm{ReLU}(Wy) \end{bmatrix}^\top \begin{bmatrix} W & 0 \\ 0 & I \end{bmatrix}^\top \begin{bmatrix} 0 & T \\ T & -2T \end{bmatrix} \begin{bmatrix} W & 0 \\ 0 & I \end{bmatrix} \begin{bmatrix} x-y \\ \mathrm{ReLU}(Wx) - \mathrm{ReLU}(Wy) \end{bmatrix}.$$

This inequality can equivalently be rewritten as

$$\rho(x - y)^\top (x - y)$$
$$- (\underbrace{Hx + G\,\mathrm{ReLU}(Wx)}_{=\mathrm{MaxMin}(x)} - Hy - G\,\mathrm{ReLU}(Wy))^\top (Hx + G\,\mathrm{ReLU}(Wx) - Hy - G\,\mathrm{ReLU}(Wy))$$
$$\geq \begin{bmatrix} Wx - Wy \\ \mathrm{ReLU}(Wx) - \mathrm{ReLU}(Wy) \end{bmatrix}^\top \begin{bmatrix} 0 & T \\ T & -2T \end{bmatrix} \begin{bmatrix} Wx - Wy \\ \mathrm{ReLU}(Wx) - \mathrm{ReLU}(Wy) \end{bmatrix},$$

which we can in turn equivalently state as

$$\rho\|x - y\|_2^2 - \|\mathrm{MaxMin}(x) - \mathrm{MaxMin}(y)\|_2^2$$
$$\geq \begin{bmatrix} Wx - Wy \\ \mathrm{ReLU}(Wx) - \mathrm{ReLU}(Wy) \end{bmatrix}^\top \begin{bmatrix} 0 & T \\ T & -2T \end{bmatrix} \begin{bmatrix} Wx - Wy \\ \mathrm{ReLU}(Wx) - \mathrm{ReLU}(Wy) \end{bmatrix} \geq 0,$$

where the last inequality holds due to $[0, 1]$-slope-restriction of ReLU, cmp. (Fazlyab et al., 2019, Lemma 1). This yields $\sqrt{\rho}\|x - y\|_2 \geq \|\mathrm{MaxMin}(x) - \mathrm{MaxMin}(y)\|_2$, which completes the proof. $\qquad\square$

## C.2 PROOF OF LEMMA 1

Let $e_i$ denote a $N$-dimensional unit vector whose $i$-th entry is 1 and all other entries are 0. Since GroupSort preserves the sum within each subgroup, we must have $e_i^\top \otimes \mathbf{1}_{n_g}^\top x = e_i^\top \otimes \mathbf{1}_{n_g}^\top \phi(x)$ for all $i = 1, \ldots, N$. This leads to the following key equality:

$$(x - y)^\top ((e_i e_i^\top) \otimes (\mathbf{1}_{n_g} \mathbf{1}_{n_g}^\top))(x - y) = (\phi(x) - \phi(y))^\top ((e_i e_i^\top) \otimes (\mathbf{1}_{n_g} \mathbf{1}_{n_g}^\top))(\phi(x) - \phi(y))$$
$$= (\phi(x) - \phi(y))^\top ((e_i e_i^\top) \otimes (\mathbf{1}_{n_g} \mathbf{1}_{n_g}^\top))(x - y) = (x - y)^\top ((e_i e_i^\top) \otimes (\mathbf{1}_{n_g} \mathbf{1}_{n_g}^\top))(\phi(x) - \phi(y)).$$

Therefore, we can introduce conic combinations of the subgroup-wise constraints and obtain the following quadratic equalities:

$$\begin{bmatrix} x - y \\ \phi(x) - \phi(y) \end{bmatrix}^\top \begin{bmatrix} \mathrm{diag}(\gamma) \otimes (\mathbf{1}_{n_g} \mathbf{1}_{n_g}^\top) & 0 \\ 0 & -\mathrm{diag}(\gamma) \otimes (\mathbf{1}_{n_g} \mathbf{1}_{n_g}^\top) \end{bmatrix} \begin{bmatrix} x - y \\ \phi(x) - \phi(y) \end{bmatrix} = 0,$$

$$\begin{bmatrix} x - y \\ \phi(x) - \phi(y) \end{bmatrix}^\top \begin{bmatrix} 0 & \mathrm{diag}(\nu) \otimes (\mathbf{1}_{n_g} \mathbf{1}_{n_g}^\top) \\ \mathrm{diag}(\nu) \otimes (\mathbf{1}_{n_g} \mathbf{1}_{n_g}^\top) & -2\,\mathrm{diag}(\nu) \otimes (\mathbf{1}_{n_g} \mathbf{1}_{n_g}^\top) \end{bmatrix} \begin{bmatrix} x - y \\ \phi(x) - \phi(y) \end{bmatrix} = 0,$$

$$\begin{bmatrix} x - y \\ \phi(x) - \phi(y) \end{bmatrix}^\top \begin{bmatrix} -2\,\mathrm{diag}(\tau) \otimes (\mathbf{1}_{n_g} \mathbf{1}_{n_g}^\top) & \mathrm{diag}(\tau) \otimes (\mathbf{1}_{n_g} \mathbf{1}_{n_g}^\top) \\ \mathrm{diag}(\tau) \otimes (\mathbf{1}_{n_g} \mathbf{1}_{n_g}^\top) & 0 \end{bmatrix} \begin{bmatrix} x - y \\ \phi(x) - \phi(y) \end{bmatrix} = 0.$$

By these equalities, (6) is equivalent to

$$\begin{bmatrix} x - y \\ \phi(x) - \phi(y) \end{bmatrix}^\top \begin{bmatrix} \mathrm{diag}(\lambda) \otimes I_{n_g} & 0 \\ 0 & -\mathrm{diag}(\lambda) \otimes I_{n_g} \end{bmatrix} \begin{bmatrix} x - y \\ \phi(x) - \phi(y) \end{bmatrix} \geq 0. \qquad \text{(C.1)}$$

We can easily see that (C.1) holds because the sorting behavior in each subgroup is 1-Lipschitz, i.e. $\|e_i^\top \otimes I_{n_g}(x - y)\|_2^2 \geq \|e_i^\top \otimes I_{n_g}(\phi(x) - \phi(y))\|_2^2$ for $i = 1, \ldots, N$. Hence, we have verified that the GroupSort/MaxMin activation satisfies the quadratic constraint (6). $\qquad\square$

## C.3 PROOF OF LEMMA 2

First, we prove that the Householder activation satisfies three key equalities on each subgroup and then we discuss the 1-Lipschitz of Householder activations. The first key equation is

$$(\phi(x) - \phi(y))^\top (I_{n_g} - vv^\top)(\phi(x) - \phi(y)) = (x - y)^\top (I_{n_g} - vv^\top)(x - y).$$

We distinguish between the cases (i) $\phi(x) = x$, $\phi(y) = y$, (ii) $\phi(x) = (I_{n_g} - 2vv^\top)x$, $\phi(y) = (I_{n_g} - 2vv^\top)y$, and (iii) $\phi(x) = x$, $\phi(y) = (I - 2vv^\top)y$ that without loss of generality corresponds to $\phi(x) = (I_{n_g} - 2vv^\top)x$, $\phi(y) = y$.

(i) $(\phi(x) - \phi(y))^\top (I_{n_g} - vv^\top)(\phi(x) - \phi(y)) = (x - y)^\top (I_{n_g} - vv^\top)(x - y)$ holds trivially.

(ii) $(\phi(x) - \phi(y))^\top (I_{n_g} - vv^\top)(\phi(x) - \phi(y))$
$= (x - y)^\top (I_{n_g} - 2vv^\top)(I_{n_g} - vv^\top)(I_{n_g} - 2vv^\top)(x - y)$
$= (x-y)^\top (I_{n_g} - vv^\top)(x-y) - 4(x-y)^\top vv^\top (I_{n_g} - vv^\top)(x-y) + 4(x-y)^\top vv^\top (I_{n_g} - vv^\top)vv^\top (x - y) = (x - y)^\top vv^\top (x - y)$ holds, using that $vv^\top (I_{n_g} - vv^\top)vv^\top = vv^\top - vv^\top = 0$.

(iii) $(\phi(x) - \phi(y))^\top (I_{n_g} - vv^\top)(\phi(x) - \phi(y))$
$= (x - (I_{n_g} - 2vv^\top)y)^\top (I_{n_g} - vv^\top)(x - (I_{n_g} - 2vv^\top)y)$
$= ((x - y) + 2vv^\top y)^\top (I_{n_g} - vv^\top)((x - y) + 2vv^\top y)$
$= (x-y)^\top (I_{n_g} - vv^\top)(x-y) - 4(x-y)^\top (I_{n_g} - vv^\top)vv^\top y + 4y^\top vv^\top (I_{n_g} - vv^\top)vv^\top y$
holds, again using that $(I_{n_g} - vv^\top)vv^\top = vv^\top - vv^\top = 0$.

Next we prove that the Householder activation satisfies the equation

$$(\phi(x) - \phi(y))^\top (I_{n_g} - vv^\top)(x - y) = (x - y)^\top (I_{n_g} - vv^\top)(x - y),$$

again considering the three cases (i), (ii) and (iii).

(i) $(\phi(x) - \phi(y))^\top (I_{n_g} - vv^\top)(x - y) = (x - y)^\top (I_{n_g} - vv^\top)(x - y)$ holds trivially.

(ii) $(\phi(x) - \phi(y))^\top (I_{n_g} - vv^\top)(x - y)$
$= (x - y)^\top (I_{n_g} - 2vv^\top)(I_{n_g} - vv^\top)(x - y)$
$= (x - y)^\top (I_{n_g} - 3vv^\top + 2vv^\top vv^\top)(x - y)$
$= (x - y)^\top (I_{n_g} - vv^\top)(x - y)$ holds.

(iii) $(\phi(x) - \phi(y))^\top (I_{n_g} - vv^\top)(x - y)$
$= (x - y^\top (I_{n_g} - 2vv^\top))(I_{n_g} - vv^\top)(x - y)$
$= (x - y)^\top (I_{n_g} - vv^\top)(x - y) - 2y^\top vv^\top (I_{n_g} - vv^\top)(x - y)$
$= (x - y)^\top (I_{n_g} - vv^\top)(x - y)$ using that $vv^\top (I_{n_g} - vv^\top) = vv^\top - vv^\top = 0$.

Accordingly, we show that the Householder activation function satisfies

$$(\phi(x) - \phi(y))^\top (I_{n_g} - vv^\top)(\phi(x) - \phi(y)) = (x - y)^\top (I_{n_g} - vv^\top)(\phi(x) - \phi(y))$$

(i) $(\phi(x) - \phi(y))^\top (I_{n_g} - vv^\top)(\phi(x) - \phi(y)) = (\phi(x) - \phi(y))^\top (I_{n_g} - vv^\top)(x - y)$ holds trivially.

(ii) $(\phi(x) - \phi(y))^\top (I_{n_g} - vv^\top)(\phi(x) - \phi(y))$
$= (x - y)^\top (I_{n_g} - 2vv^\top)(I_{n_g} - vv^\top)(\phi(x) - \phi(y))$
$= (x - y)^\top (I_{n_g} - 3vv^\top + 2vv^\top vv^\top)(\phi(x) - \phi(y))$
$= (x - y)^\top (I_{n_g} - vv^\top)(\phi(x) - \phi(y))$ holds.

(iii) $(\phi(x) - \phi(y))^\top (I_{n_g} - vv^\top)(\phi(x) - \phi(y))$
$= (x - y^\top (I_{n_g} - 2vv^\top))(I_{n_g} - vv^\top)(\phi(x) - \phi(y))$
$= (x - y)^\top (I_{n_g} - vv^\top)(x - y) - 2y^\top vv^\top (I_{n_g} - vv^\top)(\phi(x) - \phi(y))$
$= (x - y)^\top (I_{n_g} - vv^\top)(\phi(x) - \phi(y))$, using that $vv^\top (I_{n_g} - vv^\top) = vv^\top - vv^\top = 0$.

Therefore, we can introduce conic combinations of the subgroup-wise constraints and obtain the following quadratic equalities:

$$\begin{bmatrix} x - y \\ \phi(x) - \phi(y) \end{bmatrix}^\top \begin{bmatrix} \operatorname{diag}(\gamma) \otimes (\mathbf{1}_{n_g} \mathbf{1}_{n_g}^\top) & 0 \\ 0 & -\operatorname{diag}(\gamma) \otimes (\mathbf{1}_{n_g} \mathbf{1}_{n_g}^\top) \end{bmatrix} \begin{bmatrix} x - y \\ \phi(x) - \phi(y) \end{bmatrix} = 0,$$

$$\begin{bmatrix} x - y \\ \phi(x) - \phi(y) \end{bmatrix}^\top \begin{bmatrix} 0 & \operatorname{diag}(\nu) \otimes (\mathbf{1}_{n_g} \mathbf{1}_{n_g}^\top) \\ \operatorname{diag}(\nu) \otimes (\mathbf{1}_{n_g} \mathbf{1}_{n_g}^\top) & -2\operatorname{diag}(\nu) \otimes (\mathbf{1}_{n_g} \mathbf{1}_{n_g}^\top) \end{bmatrix} \begin{bmatrix} x - y \\ \phi(x) - \phi(y) \end{bmatrix} = 0,$$

$$\begin{bmatrix} x - y \\ \phi(x) - \phi(y) \end{bmatrix}^\top \begin{bmatrix} -2\operatorname{diag}(\tau) \otimes (\mathbf{1}_{n_g} \mathbf{1}_{n_g}^\top) & \operatorname{diag}(\tau) \otimes (\mathbf{1}_{n_g} \mathbf{1}_{n_g}^\top) \\ \operatorname{diag}(\tau) \otimes (\mathbf{1}_{n_g} \mathbf{1}_{n_g}^\top) & 0 \end{bmatrix} \begin{bmatrix} x - y \\ \phi(x) - \phi(y) \end{bmatrix} = 0.$$

By these equalities, (6) is equivalent to

$$\begin{bmatrix} x-y \\ \phi(x)-\phi(y) \end{bmatrix}^\top \begin{bmatrix} \mathrm{diag}(\lambda)\otimes I_{n_g} & 0 \\ 0 & -\,\mathrm{diag}(\lambda)\otimes I_{n_g} \end{bmatrix} \begin{bmatrix} x-y \\ \phi(x)-\phi(y) \end{bmatrix} \geq 0. \tag{C.2}$$

The Householder activation is gradient norm preserving by design as $I$ and $I-2vv^\top$ have only singular values 1. Accordingly, it is 1-Lipschitz, i.e., $\|x-y\|_2 \geq \|\phi(x)-\phi(y)\|_2$ holds and inequality (C.2) is a conic combination of the Lipschitz property. $\qquad\square$

## C.4 Proof of Theorem 1

We want to prove that (9) implies $\|f_\theta(x)-f_\theta(y)\|_2^2 \leq \rho\|x-y\|_2^2$ for any $x,y \in \mathbb{R}^{n_0}$. To do this, we set $x^0 = x$, and $y^0 = y$. Then we define

$$\mathbf{x} = \begin{bmatrix} x^0 \\ x^1 \\ \vdots \\ x^{l-1} \end{bmatrix}, \quad \mathbf{y} = \begin{bmatrix} y^0 \\ y^1 \\ \vdots \\ y^{l-1} \end{bmatrix} \tag{C.3}$$

where $x^i = \phi(W_i x^{i-1} + b_i)$ and $y^i = \phi(W_i y^{i-1} + b_i)$. Now we left and right multiply (9) with $(\mathbf{x}-\mathbf{y})^\top$ and its transpose, respectively. This leads to

$$(\mathbf{x}-\mathbf{y})^\top \begin{bmatrix} A \\ B \end{bmatrix}^\top \begin{bmatrix} T-2S & P+S \\ P+S & -T-2P \end{bmatrix} \begin{bmatrix} A \\ B \end{bmatrix} (\mathbf{x}-\mathbf{y})$$

$$+(\mathbf{x}-\mathbf{y})^\top \begin{bmatrix} -\rho I_{n_0} & 0 & \dots & 0 \\ 0 & 0 & \dots & 0 \\ \vdots & \vdots & \ddots & \vdots \\ 0 & 0 & \dots & W_l^\top W_l \end{bmatrix} (\mathbf{x}-\mathbf{y}) \leq 0,$$

which can be rewritten as

$$\rho(x^0-y^0)^\top(x^0-y^0) - (x^{l-1}-y^{l-1})^\top W_l^\top W_l (x^{l-1}-y^{l-1})$$
$$\geq \begin{bmatrix} A\mathbf{x}-A\mathbf{y} \\ \phi(A\mathbf{x}+b)-\phi(A\mathbf{y}+b) \end{bmatrix}^\top \begin{bmatrix} T-2S & P+S \\ P+S & -T-2P \end{bmatrix} \begin{bmatrix} A\mathbf{x}-A\mathbf{y} \\ \phi(A\mathbf{x}+b)-\phi(A\mathbf{y}+b) \end{bmatrix}, \tag{C.4}$$

where $b^\top = \begin{bmatrix} b_1^\top & b_2^\top & \dots & b_{l-1}^\top \end{bmatrix}$. Based on Lemma 1 (Lemma 2), the right side of (C.4) is guaranteed to be non-negative. Therefore, we have

$$\rho(x^0-y^0)^\top(x^0-y^0) - (x^{l-1}-y^{l-1})^\top W_l^\top W_l (x^{l-1}-y^{l-1}) \geq 0.$$

Noting that $f_\theta(x)-f_\theta(y) = W_l x^{l-1} + b_l - W_l y^{l-1} - b_l = W_l(x^{l-1}-y^{l-1})$ and $x^0 - y^0 = x-y$, we finally arrive at

$$\|f_\theta(x)-f_\theta(y)\|_2^2 \leq \rho\|x-y\|_2^2.$$

This completes the proof. $\qquad\square$

## C.5 Proof of Theorem 2

This theorem considers a single-layer neural network $f_\theta(x) = W_2\phi(W_1 x + b_1) + b_2$ with $f_\theta(x) \in \mathbb{R}$. Given $x^0, y^0 \in \mathbb{R}^{n_0}$, we set $x^1 = \phi(W_1 x^0 + b_1)$, and $y^1 = \phi(W_1 y^0 + b_1)$. We left and right multiply (14) with $\begin{bmatrix} 1 & (x^0-y^0)^\top & (x^1-y^1)^\top \end{bmatrix}$ and its transpose, respectively, and obtain

$$\begin{bmatrix} 1 \\ x^0-y^0 \\ x^1-y^1 \end{bmatrix}^\top \begin{bmatrix} \mathbf{1}_{n_0}^\top \mu - 2\rho & 0 & W_2 \\ 0 & W_1^\top(T-2S)W_1 - Q & W_1^\top(P+S) \\ W_2^\top & (P+S)W_1 & -T-2P \end{bmatrix} \begin{bmatrix} 1 \\ x^0-y^0 \\ x^1-y^1 \end{bmatrix} \leq 0, \tag{C.5}$$

which is equivalent to

$$2\rho - 2W_2(x^1-y^1) \geq \sum_{i=1}^{n_0} \mu_i(1-\delta_i^2) +$$

$$\begin{bmatrix} W_1(x^0-y^0) \\ \phi(W_1 x_0 + b_1)-\phi(W_1 y_0 + b_1) \end{bmatrix}^\top \begin{bmatrix} T-2S & P+S \\ P+S & -T-2P \end{bmatrix} \begin{bmatrix} W_1(x^0-y^0) \\ \phi(W_1 x^0 + b_1)-\phi(W_1 y^0 + b_1) \end{bmatrix}, \tag{C.6}$$

where $\delta_i$ is the $i$-th entry of the vector $(x^0 - y^0)$. Given that we aim to find some $\rho$ that solves problem (13), we consider only pairs of $x^0, y^0, x^1, y^1$ that satisfy the constraints in (13). Based on these constraints, the right hand side of (C.6) has to be non-negative. This means that we must have $\rho \geq W_2(x^1 - y^1)$. Therefore, $\rho$ provides an upper bound for the objective of problem (13), and we have $\rho \geq L_{\min}$. This immediately leads to the desired conclusion. $\qquad\square$

### C.6 PROOF OF THEOREM 3

Let $(\mathbf{x}, \mathbf{y})$ be defined by (C.3). We left and right multiply (15) with $\begin{bmatrix} 1 & (\mathbf{x} - \mathbf{y})^\top \end{bmatrix}$ and its transpose, respectively, and obtain

$$(\mathbf{x} - \mathbf{y})^\top \begin{bmatrix} A \\ B \end{bmatrix}^\top \begin{bmatrix} T - 2S & P + S \\ P + S & -T - 2P \end{bmatrix} \begin{bmatrix} A \\ B \end{bmatrix} (\mathbf{x} - \mathbf{y})$$

$$+ \begin{bmatrix} 1 \\ (\mathbf{x} - \mathbf{y}) \end{bmatrix}^\top \begin{bmatrix} \mathbf{1}_{n_0}^\top \mu - 2\rho & 0 & \dots & W_l \\ 0 & -Q & \dots & 0 \\ \vdots & \vdots & \ddots & \vdots \\ W_l^\top & 0 & \dots & 0 \end{bmatrix} \begin{bmatrix} 1 \\ (\mathbf{x} - \mathbf{y}) \end{bmatrix} \leq 0,$$

which is equivalent to

$$2\rho - 2W_l(x^{l-1} - y^{l-1}) \geq \sum_{i=1}^{n_0} \mu_i(1 - \delta_i^2) +$$

$$\begin{bmatrix} A\mathbf{x} - A\mathbf{y} \\ \phi(A\mathbf{x} + b) - \phi(A\mathbf{y} + b) \end{bmatrix}^\top \begin{bmatrix} T - 2S & P + S \\ P + S & -T - 2P \end{bmatrix} \begin{bmatrix} A\mathbf{x} - A\mathbf{y} \\ \phi(A\mathbf{x} + b) - \phi(A\mathbf{y} + b) \end{bmatrix}, \tag{C.7}$$

where $b^\top = \begin{bmatrix} b_1^\top & b_2^\top & \dots & b_{l-1}^\top \end{bmatrix}$, and $\delta_i$ is the $i$-th entry of the vector $(x^0 - y^0)$. Following the same arguments in Section C.5, we can show $\rho \geq W_l(x^{l-1} - y^{l-1})$, which leads to the desired conclusion. $\qquad\square$

### C.7 PROOF OF THEOREM 4

Given $x^0 \in \mathbb{R}^{n_0}$, let $\{x^i\}_{i=1}^l$ be defined by (16). Similarly, given $y^0 \in \mathbb{R}^{n_0}$, we can calculate the output of the multi-layer residual network based on the recursion $y^i = y^{i-1} + G_i\phi(W_iy^{i-1} + b_i)$.

We define $\tilde{x}^i = \phi(W_ix^{i-1} + b_i)$ for $i = 1, 2, \dots, l$. Then we have $x^i = x^{i-1} + G_i\tilde{x}^i$ for $i = 1, 2, \dots, l$. Via setting $\tilde{x}^0 = x^0$, we can unroll this recursive expression to obtain the following relationship:

$$x^i = \tilde{x}^0 + \sum_{k=1}^i G_k\tilde{x}^k.$$

Therefore, we have $\tilde{x}^i = \phi(W_i(\tilde{x}^0 + \sum_{k=1}^{i-1} G_k\tilde{x}^k) + b_i)$ for $i = 1, \dots, l$.

Similarly, we define $\tilde{y}^i = \phi(W_iy^{i-1} + b_i)$. We must have $\tilde{y}^i = \phi(W_i(\tilde{y}^0 + \sum_{k=1}^{i-1} G_k\tilde{y}^k) + b_i)$ for $i = 1, \dots, l$.

Next, we define

$$\tilde{\mathbf{x}} = \begin{bmatrix} \tilde{x}^0 \\ \tilde{x}^1 \\ \vdots \\ \tilde{x}^l \end{bmatrix}, \quad \tilde{\mathbf{y}} = \begin{bmatrix} \tilde{y}^0 \\ \tilde{y}^1 \\ \vdots \\ \tilde{y}^l \end{bmatrix}, \quad b = \begin{bmatrix} b_1 \\ b_2 \\ \vdots \\ b_l \end{bmatrix} \tag{C.8}$$

Then it is straightforward to verify $\tilde{B}\tilde{\mathbf{x}} = \phi(\tilde{A}\tilde{\mathbf{x}} + b)$ and $\tilde{B}\tilde{\mathbf{y}} = \phi(\tilde{A}\tilde{\mathbf{y}} + b)$. This leads to

$$\tilde{B}(\tilde{\mathbf{x}} - \tilde{\mathbf{y}}) = \phi(\tilde{A}\tilde{\mathbf{x}} + b) - \phi(\tilde{A}\tilde{\mathbf{y}} + b) \tag{C.9}$$

Now we left and right multiply (17) with $(\tilde{\mathbf{x}} - \tilde{\mathbf{y}})^\top$ and its transpose, respectively, and obtain

$$
(\tilde{\mathbf{x}} - \tilde{\mathbf{y}})^\top \begin{bmatrix} \tilde{A} \\ \tilde{B} \end{bmatrix}^\top \begin{bmatrix} T - 2S & P + S \\ P + S & -T - 2P \end{bmatrix} \begin{bmatrix} \tilde{A} \\ \tilde{B} \end{bmatrix} (\tilde{\mathbf{x}} - \tilde{\mathbf{y}})
$$

$$
+ (\tilde{\mathbf{x}} - \tilde{\mathbf{y}})^\top \begin{bmatrix} (1-\rho)I_{n_0} & G_1 & \dots & G_l \\ G_1^\top & G_1^\top G_1 & \dots & G_1^\top G_l \\ \vdots & \vdots & \ddots & \vdots \\ G_l^\top & G_l^\top G_1 & \dots & G_l^\top G_l \end{bmatrix} (\tilde{\mathbf{x}} - \tilde{\mathbf{y}}) \le 0,
$$

which can be combined with (C.9) to show

$$
\rho \|\tilde{x}^0 - \tilde{y}^0\|_2^2 - \|\tilde{x}^0 - \tilde{y}^0 + \sum_{k=1}^l G_k(\tilde{x}^k - \tilde{y}^k)\|_2^2 \tag{C.10}
$$

$$
\ge \begin{bmatrix} \tilde{A}\tilde{\mathbf{x}} - \tilde{A}\tilde{\mathbf{y}} \\ \phi(\tilde{A}\tilde{\mathbf{x}} + b) - \phi(\tilde{A}\tilde{\mathbf{y}} + b) \end{bmatrix} \begin{bmatrix} T - 2S & P + S \\ P + S & -T - 2P \end{bmatrix} \begin{bmatrix} \tilde{A}\tilde{\mathbf{x}} - \tilde{A}\tilde{\mathbf{y}} \\ \phi(\tilde{A}\tilde{\mathbf{x}} + b) - \phi(\tilde{A}\tilde{\mathbf{y}} + b) \end{bmatrix}.
$$

Based on Lemma 1/Lemma 2, we know the right side of (C.10) must be non-negative. Therefore, we have

$$
\|\tilde{x}^0 - \tilde{y}^0 + \sum_{k=1}^l G_k(\tilde{x}^k - \tilde{y}^k)\|_2 \le \sqrt{\rho}\|\tilde{x}^0 - \tilde{y}^0\|_2.
$$

Recall that we have $x - y = \tilde{x}^0 - \tilde{y}^0$ and $f_\theta(x) - f_\theta(y) = \tilde{x}^0 - \tilde{y}^0 + \sum_{k=1}^l G_k(\tilde{x}^k - \tilde{y}^k)$. Hence we have $\|f_\theta(x) - f_\theta(y)\|_2 \le \sqrt{\rho}\|x - y\|_2$, which completes the proof. $\square$

# D  IMPLEMENTATION DETAILS FOR IMPROVED SCALABILITY OF THE SDP

It is well-known that SDPs run into scalability problems for large-scale problems. To improve the scalability of the proposed SDPs for Lipschitz constant estimation, efforts have been made to exploit the structure of the matrix that characterizes the SDP condition, using chordal sparsity (Newton & Papachristodoulou, 2023; Xue et al., 2022) or layer-by-layer interpretations (Pauli et al., 2023a). We incorporated these tricks in our implementations. In the following, we present details on our practical implementations of the SDPs based on Theorem 1 and Theorem 3 and address some additional tricks to trade off efficiency and accuracy of the SDP.

## D.1  PRACTICAL IMPLEMENTATION FOR SDP BASED ON THEOREM 1

To enhance computational tractability in large-scale problems, it is desirable to formulate layer-wise SDP conditions rather than the sparse end-to-end one (Pauli et al., 2023a; Xue et al., 2022). To make such simplifications, we set $\nu = 0$ and $\tau = 0$, which does not change the final solution of the above SDP for feedforward neural networks in (1). Therefore, we can just set $S = P = 0$. Further, we denote $T_i = \lambda_i I_{n_g} + \gamma_i (\mathbf{1}_{n_g} \mathbf{1}_{n_g}^\top)$, which yields $T = \mathrm{blkdiag}(T_1, \dots, T_{l-1})$. Then, (9) can equivalently be stated as

$$
\mathrm{blkdiag}(W_1^\top T_1 W_1 - \rho I, W_2^\top T_2 W_2 - T_1, \dots, W_l^\top W_l - T_{l-1}) \preceq 0. \tag{D.1}
$$

Consequently, we can implement a set of $l$ SDP constraints, using the blocks in the matrix in (D.1), rather than using the large and sparse condition (9). This idea follows along the lines of Newton & Papachristodoulou (2023); Xue et al. (2022), where the sparsity pattern of the block-tridiagonal LMI in Fazlyab et al. (2019) is exploited. The decomposition of the matrix in (D.1) is straightforward due to its even simpler block-diagonal structure. Specifically, setting $T_0 = \rho I$, we solve the SDP

$$
\min_{\rho, T} \rho \quad \text{s.t.} \quad W_1^\top T_1 W_1 - T_0 \preceq 0, \; W_2^\top T_2 W_2 - T_1 \preceq 0, \; \dots, \; W_{l-1}^\top W_{l-1} - T_{l-1} \preceq 0. \tag{D.2}
$$

## D.2  PRACTICAL IMPLEMENTATION FOR SDP BASED ON THEOREM 3

We argue accordingly for $\ell_\infty$ Lipschitz bounds based on Theorem 3. Minimizing $\rho$ subject to (15) yields a SDP problem. For large-scale implementations, we set $S = P = 0$, which reduces the

number of decision variables without affecting the solution. Further, we reorder rows and columns of the matrix in (15) by a similarity transformation. The rearranged version of the matrix in (15) has a block-diagonal structure that leads to the following decoupled conditions with $T_0 = Q$:

$$T_{i-1} - W_i^\top T_i W_i \succeq 0, \; i = 1, \ldots, l-1, \quad \begin{bmatrix} T_{l-1} & W_l^\top \\ W_l & 2\rho - \sum_{j=1}^{n_0} \mu_j \end{bmatrix} \succeq 0. \tag{D.3}$$

This yields a set of non-sparse constraints such that the resulting SDP can be solved in a more efficient way.

### D.3 TRADE-OFF BETWEEN ACCURACY AND COMPUTATIONAL EFFICIENCY

In Section 5, we focus on the computation times of the SDP-based methods. For completeness and to discuss the trade-off between accuracy and computational efficiency, we include computation times for the naive $\ell_2$ bounds FGL in this section. This involves the lower bound from sampling and the matrix product bound for all models analyzed in Table 1.

Table 2: This table presents the computation times in seconds of all naive $\ell_2$ bounds for the neural networks used in Table 1: the sample bound evaluated on 200k samples, the FGL bound, and the matrix product bound. See Section 5 for details on the bounds.

| Model (units per layer) | | $\ell_2$ | | |
|---|---|---|---|---|
| | | Sample | FGL | MP |
| 2-layer | 16 units | 12.02 | 4.54 | 6.16 |
| | 32 units | 13.08 | 81.92 | 5.33 |
| | 64 units | 12.84 | - | 3.95 |
| | 128 units | 12.61 | - | 3.6 |
| 5-layer | 32 units | 13.18 | - | 5.43 |
| | 64 units | 14.83 | - | 5.68 |
| 8-layer | 32 units | 14.14 | - | 5.57 |
| | 64 units | 14.27 | - | 5.6 |
| | 128 units | 14.15 | - | 9.85 |
| | 256 units | 13.74 | - | 9.3 |
| 18-layer | 32 units | 23.28 | - | 7.64 |
| | 64 units | 21.09 | - | 9.22 |
| | 128 units | 19.48 | - | 4.9 |

The computation of the matrix product bound (MP), while naive, is quite fast, taking less than 10 s for all our models. It provides reasonably accurate bounds when the number of layers is small, but becomes very loose for deep neural networks. LipSDP-NSR is in comparison computationally more expensive, while providing significantly tighter bounds, cmp. Table 1. The computation of a lower bound by sampling (Sample) takes between 12 and 23 s on our models, which is also comparably cheap. This bound is not helpful in further verification steps but it gives us an idea on the range in which the true Lipschitz constant lies. Finally, it may seem surprising at first that the computation of FGL can be computed comparably fast to LipSDP-NSR for the small models 2FC-16 and 2FC-32. However, due to the exponential complexity of this approach it becomes intractable for all larger models.

### D.4 FURTHER TRICKS TO TRADE OFF EFFICIENCY AND ACCURACY OF THE SDP

Considering less decision variables per layer, cmp. the variants LipSDP-Neuron and LipSDP-Layer in Fazlyab et al. (2019), can be used to trade off efficiency and accuracy of the SDP.

- **LipSDP-NSR-Neuron-2**: 2 decision variables per group (as implemented in problem (D.2))

- **LipSDP-NSR-Neuron-1**: 1 decision variable per group ($\gamma_i = 0, i = 1, \ldots, l$)

Table 3: This table presents additional results of our LipSDP-NSR method on several feed-forward neural networks trained on MNIST with more than 96% accuracy. The first table describe the result with classic training while the second table describes the results for $\ell_\infty$ adversarial training. We use the same setting as described in the paper.

| Model (units per layer) | | $\ell_2$ | | | | $\ell_\infty$ | | | | | |
|---|---|---|---|---|---|---|---|---|---|---|---|
| | | Sample | FGL | LipSDP-NSR | | MP | Sample | FGL | LipSDP-NSR | | Norm Eq. | MP |
| **Classic Training** | | | | | | | | | | | | |
| 2-layer | 16 units | 21.18 | 21.18 | 21.39 | (76) | 22.86 | 147.63 | 162.14 | 203.98 | (149) | 599.04 | 488.85 |
| | 32 units | 20.40 | 20.40 | 21.09 | (152) | 23.65 | 196.21 | 202.78 | 244.14 | (220) | 590.43 | 782.25 |
| | 64 units | 17.51 | - | 20.34 | (279) | 22.67 | 212.23 | - | 300.21 | (425) | 569.58 | 1164.53 |
| | 128 units | 16.25 | - | 19.07 | (673) | 21.94 | 179.04 | - | 297.00 | (650) | 533.86 | 1339.68 |
| 5-layer | 32 units | 107.46 | - | 176.69 | (130) | 240.15 | 1073.68 | - | 2471.64 | (271) | 4947.20 | 2.96e+05 |
| | 64 units | 101.04 | - | 132.48 | (359) | 166.23 | 570.29 | - | 1688.85 | (581) | 3709.43 | 2.63e+05 |
| 8-layer | 32 units | 118.03 | - | 414.02 | (188) | 711.15 | 1353.52 | - | 5134.45 | (286) | 11592.60 | 1.36e+07 |
| | 64 units | 93.33 | - | 246.35 | (357) | 405.13 | 863.88 | - | 3620.96 | (494) | 6897.82 | 3.60e+07 |
| | 128 units | 81.99 | - | 203.65 | (595) | 306.68 | 701.12 | - | 2516.24 | (912) | 5702.34 | 1.01e+08 |
| | 256 units | 54.83 | - | 149.90 | (1337) | 225.18 | 425.83 | - | 2133.17 | (2256) | 4197.23 | 4.22e+08 |
| 18-layer | 32 units | 288.22 | - | 1.45e+04 | (270) | 7.70e+04 | 2335.92 | - | 1.52e+05 | (285) | 4.06e+05 | 2.88e+13 |
| | 64 units | 361.36 | - | 4.77e+04 | (496) | 2.05e+05 | 1677.43 | - | 7.13e+05 | (559) | 1.34e+06 | 4.03e+15 |
| | 128 units | 150.60 | - | 5.56e+05 | (1047) | 8.73e+06 | 1457.68 | - | 5.70e+06 | (1331) | 1.56e+07 | 9.35e+17 |
| **$\ell_\infty$ Adversarial training** | | | | | | | | | | | | |
| 2-layer | 16 units | 13.30 | 14.79 | 15.04 | (76) | 17.47 | 57.61 | 65.69 | 92.90 | (179) | 421.12 | 258.70 |
| | 32 units | 14.51 | 15.73 | 15.88 | (120) | 17.86 | 98.30 | 106.28 | 148.03 | (242) | 444.64 | 453.64 |
| | 64 units | 20.31 | - | 25.32 | (255) | 28.95 | 141.66 | - | 229.96 | (423) | 708.96 | 900.19 |
| | 128 units | 31.90 | - | 44.48 | (559) | 52.19 | 307.72 | - | 485.39 | (802) | 1245.44 | 2287.62 |
| 5-layer | 32 units | 24.02 | - | 62.62 | (138) | 93.75 | 162.47 | - | 632.41 | (294) | 1753.36 | 82751.08 |
| | 64 units | 68.40 | - | 64.06 | (138) | 221.65 | 591.78 | - | 737.41 | (291) | 1793.68 | 596213.27 |
| 8-layer | 32 units | 23.55 | - | 134.44 | (155) | 260.87 | 178.88 | - | 1294.9 | (310) | 3764.32 | 9148849 |
| | 64 units | 67.43 | - | 284.41 | (333) | 531.62 | 546.66 | - | 3209.0 | (621) | 7963.48 | 7.48E+07 |
| | 128 units | 123.39 | - | 461.46 | (624) | 814.95 | 1073.40 | - | 5309.5 | (1096) | 12920.88 | 3.60E+08 |
| | 256 units | 176.81 | - | 694.33 | (1835) | 1132.61 | 1743.15 | - | 9415.6 | (3578) | 19441.24 | 1.66E+09 |
| 18-layer | 32 units | 244.79 | - | 9392.8 | (182) | 37785.55 | 1508.21 | - | 80042 | (300) | 262998 | 1.54E+13 |
| | 64 units | 210.53 | - | 10618 | (428) | 45570.58 | 1396.26 | - | 87044 | (565) | 297304 | 5.20E+14 |
| | 128 units | 380.77 | - | 15440 | (922) | 66761.66 | 2194.38 | - | 1.6205E+05 | (1117) | 432320 | 3.63E+16 |

- **LipSDP-NSR-Layer-2**: 2 decision variables per layer ($\lambda_1 = \lambda_2 = \cdots = \lambda_l$, $\gamma_1 = \gamma_2 = \cdots = \gamma_N$)
- **LipSDP-NSR-Layer-1**: 1 decision variable per layer ($\lambda_1 = \lambda_2 = \cdots = \lambda_l$, $\gamma_1 = \gamma_2 = \cdots = \gamma_N = 0$, equivalent to norm of weights)

Another trick is to split a deep neural network in subnetworks. The product of the Lipschitz constants of the subnetworks yields a Lipschitz bound for the entire neural network.

# E   ADDITIONAL NUMERICAL RESULTS AND DISCUSSIONS

In Table 3, we present additional results on neural networks with the same architectures as in 1 that were also trained on the MNIST dataset. The upper half of Table 3 states results from classic training and the lower half from $\ell_\infty$ adversarial training.

## E.1   MORE DISCUSSIONS ON THE NUMERICAL RESULTS IN THE MAIN PAPER

For all the models in Tables 1 and 3, we observe that LipSDP-NSR provides tighter Lipschitz bounds than the norm product of the weight matrices. For small neural networks, e.g., the 2-layer network with 16 units, the deviation of LipSDP-NSR to the true Lipschitz constant under the $\ell_2$ perturbation is especially small. We note that the computation time increases in polynomial time with the number of units and decision variables. Due to the convenient block-wise implementation of the SDP conditions, we observe good scalability of LipSDP-NSR to neural networks with an increasing number of layers, being able to handle 18-layer neural networks. In comparison to LipSDP-RR, LipSDP-NSR shows tighter bounds especially for larger neural networks and its computational advantages also become apparent for larger neural networks. Table 1 shows that for small neural networks LipSDP-RR provides comparable bounds to LipSDP-NSR and is even evaluated faster. But for larger neural networks, the bounds of LipSDP-RR are much looser and the computation time blows up. For all models, the computation time required to solve the SDP for the $\ell_2$ bounds is slightly faster than the $\ell_\infty$ case. This is expected as the constraints in (D.3) include a larger SDP condition for the last layer

than problem (D.2) and additional decision variables $\mu$. We note that our $\ell_\infty$ bounds are not only significantly smaller than the $\ell_\infty$ bounds of the matrix norm product but also notably smaller (roughly by factor 2) than the bounds computed via $\ell_2$ Lipschitz bounds and the equivalence of norms.

### E.2 EXTRA NUMERICAL RESULTS ON RESIDUAL NEURAL NETWORKS

In this section, we present numerical results of LipSDP-NSR on residual neural networks as introduced in Section 4.4. The first and the last layer of our $l$-layer residual neural networks are linear layers and the fully residual part consists of the remaining $l - 2$ layers. In our examples, we choose $G_i = I$ in all residual layers. We compare our bounds to a sampling-based lower bound, and the matrix product, where for the $i$-th residual layer the Lipschitz constant is bounded by $1 + \|G_i\|_2\|W_i\|_2$. To compute LipSDP-NSR, we analyze the fully residual part of the neural network using an SDP based on (4). Then we multiply the result with $\|W_1\|\|W_l\|$ to obtain a bound for the entire network. We state the results for the choice $P \neq 0$ and $S = 0$, as well as for $P = S = 0$ in (17).

Our results are summarized in Table 4. We observe that LipSDP-NSR provides more accurate Lipschitz bounds than the matrix product. The SDP condition (17) is non-sparse and in this case, it is non-trivial to break down the constraint (17) into multiple smaller constraints. This is reflected in the computation time of the SDP solution. The computation times for small residual networks are lower than for small feed-forward networks, but they increase more drastically for larger residual neural networks. Comparing the results for $P \neq 0$, $S = 0$ and the ones for $P = S = 0$, respectively, in (17), we notice that the computation times are lower for the reduced number of decision variables ($P = S = 0$), yet the accuracy of the Lipschitz bound is worse for $P = S = 0$.

Table 4: This table presents the results of our LipSDP-NSR method on several residual neural networks trained on MNIST with more than 90% accuracy. We compare against several metrics which are described in detail in Section 5. For LipSDP-NSR, we state results choosing $P \neq 0$ and $S = 0$ as well as choosing $P = S = 0$ in (17).

| Model (units per layer) | | $\ell_2$ | | | | |
|---|---|---|---|---|---|---|
| | | **Sample** | **LipSDP-NSR** | | | **MP** |
| | | | $P \neq 0,\ S = 0$ | | $P = S = 0$ | | |
| 5-layer | 32 units | 15.33 | 29.75 | (2) | 33.04 | (1) | 55.64 |
| | 64 units | 14.28 | 31.40 | (21) | 34.64 | (11) | 53.25 |
| 8-layer | 32 units | 8.99 | 21.34 | (18) | 22.98 | (7) | 46.60 |
| | 64 units | 8.34 | 17.88 | (371) | 18.52 | (115) | 35.76 |
| | 128 units | 7.82 | 17.68 | (4629) | 18.14 | (2112) | 34.90 |

## F ADDITIONAL NETWORK ARCHITECTURES

### F.1 SINGLE-LAYER RESIDUAL NETWORK WITH GROUPSORT/HOUSEHOLDER ACTIVATIONS

For completeness, we address the special case of a single-layer residual neural network. Consider a single-layer residual network model

$$f_\theta(x) = H_1 x + G_1 \phi(W_1 x + b_1). \tag{F.1}$$

If $\phi$ is 1-Lipschitz, one will typically use a triangle inequality to obtain the Lipschitz bound $\|H_1\|_2 + \|G_1\|_2\|W_1\|_2$. However, this bound can be quite loose. If $\phi$ is slope-restricted on $[0, 1]$, a variant of LipSDP (Araujo et al., 2023, Theorem 4) provides improved Lipschitz bounds. Similarly, if $\phi$ is GroupSort/Householder, we can apply Lemma 1 to generalize (Araujo et al., 2023, Theorem 4) as follows.

**Theorem F.1.** *Consider a single-layer generalized residual network (F.1), where $\phi : \mathbb{R}^{n_1} \to \mathbb{R}^{n_1}$ is GroupSort (Householder) with group size $n_g$. Denote $N = \frac{n_1}{n_g}$. Suppose there exist $\rho > 0$,*

$\gamma, \nu, \tau \in \mathbb{R}^N$, $\lambda \in \mathbb{R}^N_+$, *such that*

$$\begin{bmatrix} I & 0 \\ H_1 & G_1 \end{bmatrix}^\top \begin{bmatrix} \rho I & 0 \\ 0 & -I \end{bmatrix} \begin{bmatrix} I & 0 \\ H_1 & G_1 \end{bmatrix} \succeq \begin{bmatrix} W_1^\top (T - 2S) W_1 & W_1^\top (P + S) \\ (P + S) W_1 & -T - 2P \end{bmatrix}, \tag{F.2}$$

*where $(T, S, P)$ are given by equation (7) (equation (8)). Then $f_\theta$ is $\sqrt{\rho}$-Lipschitz in the $\ell_2 \to \ell_2$ sense.*

*Proof.* Given $x^0, y^0 \in \mathbb{R}^{n_0}$, we set $x^1 = \phi(W_1 x^0 + b_1)$, and $y^1 = \phi(W_1 y^0 + b_1)$. We left and right multiply (F.2) by $\begin{bmatrix} (x^0 - y^0)^\top & (x^1 - y^1)^\top \end{bmatrix}$ and its transpose respectively, and obtain

$$\rho \| x^0 - y^0 \|_2^2 - \| H_1 (x^0 - y^0) + G_1 (x^1 - y^1) \|_2^2$$
$$\geq \begin{bmatrix} W_1 (x^0 - y^0) \\ \phi(W_1 x^0 + b_1) - \phi(W_1 y^0 + b_1) \end{bmatrix}^\top \begin{bmatrix} T - 2S & P + S \\ P + S & -T - 2P \end{bmatrix} \begin{bmatrix} W_1 (x^0 - y^0) \\ \phi(W_1 x^0 + b_1) - \phi(W_1 y^0 + b_1) \end{bmatrix}. \tag{F.3}$$

Based on Lemma 1, we know the left side of (F.3) is non-negative. Therefore, the left side of (F.3) is also non-negative. This immediately leads to the desired input-output bound. $\square$

## F.2 Convolutional neural networks with GroupSort/Householder activations

All results for fully connected neural networks in Section 4 can also be applied to convolutional neural networks, unrolling the convolutional layers as fully connected matrices based on Toeplitz matrices. For efficiency and scalability, SDP-based Lipschitz constant estimation has been extended to exploit the structure in convolutional neural networks (CNNs) (Pauli et al., 2023a; Gramlich et al., 2023b). In particular, they use a state space representation for convolutions that enables the formulation of compact SDP conditions. Combining those results with our newly derived quadratic constraint (6), we can extend our Lipschitz analysis to address larger-scale convolutional neural networks with GroupSort/Householder activation functions. In this section, we briefly streamline this extension and refer the interested reader to Gramlich et al. (2023b) for details.

Similar to the previous case where we modify the choice of the matrix $X$ in the original LipSDP constraint (10) to derive LipSDP-NSR, cmp. Section 3, we can adapt the SDP condition in Gramlich et al. (2023b) to address CNNs with GroupSort/Householder activation functions via modifying a corresponding matrix using Lemma 1 or Lemma 2. Specifically, setting $(Q^w, S^w, R^w)$ in (Gramlich et al., 2023b, Theorem 5) as $Q^w = T - 2S$, $S^w = P + S$, and $R^w = -T - 2P$ with $(T, S, P)$ given by (7)/(8) immediately leads to the modified SDP condition for Lipschitz constant estimation of CNNs with GroupSort/Householder activation functions.

## F.3 Deep equilibrium models with GroupSort/Householder activations

In contrast to explicit feed-forward networks, we may also consider implicit learning models. In this section, we consider the so-called deep-equilibrium models (DEQ) (Bai et al., 2019). Given an input $x \in \mathbb{R}^n$, the DEQ output is obtained by solving a nonlinear equation for the hidden variable $z \in \mathbb{R}^d$:

$$z = \phi(W z + U x + b_z), \quad y = W_o z + b_y, \tag{F.4}$$

where $W \in \mathbb{R}^{d \times d}$, $U \in \mathbb{R}^{d \times n}$, $W_o \in \mathbb{R}^{m \times d}$, $b_z \in \mathbb{R}^d$, and $b_y \in \mathbb{R}^m$ are the model parameters to be trained. Previously, SDP-based Lipschitz bounds for DEQs have been obtained for the case where $\phi$ is slope-restricted on $[0, 1]$ (Revay et al., 2020). Here we will extend these results to the case where $\phi$ is a GroupSort/Householder activation function.

The DEQ model (F.4) is well-posed if for each input $x \in \mathbb{R}^m$, there exists a unique solution $z \in \mathbb{R}^d$ satisfying (F.4). First, we leverage the incremental quadratic constraint in Lemma 1 to derive a sufficient condition that ensures the DEQ (F.4) to be well-posed.

**Theorem F.2.** *Consider the DEQ (F.4), where $\phi : \mathbb{R}^d \to \mathbb{R}^d$ is GroupSort (Householder) with group size $n_g$. Denote $N = \frac{d}{n_g}$. If there exist $\lambda \in \mathbb{R}^N_+$, $\gamma, \nu, \tau \in \mathbb{R}^N$ and $\Pi \succ 0$ such that*

$$\begin{bmatrix} -2\Pi & \Pi \\ \Pi & 0 \end{bmatrix} + \begin{bmatrix} W^\top (T - 2S) W & W^\top (P + S) \\ (P + S) W & -T - 2P \end{bmatrix} \prec 0, \tag{F.5}$$

*where $(T, S, P)$ are defined by equation (7) (equation (8)), then the DEQ (F.4) is well-posed.*

*Proof.* If we can show that the following ODE with any fixed $x \in \mathbb{R}^n$ has a unique equilibrium point

$$\frac{\mathrm{d}}{\mathrm{d}t} z(t) = -z(t) + \phi(Wz(t) + Ux + b_z), \tag{F.6}$$

then (F.4) is guaranteed to be well-posed. We will show that the condition F.5 guarantees that the ODE (F.6) converges globally to a unique equilibrium point via standard contraction arguments. Obviously, the GroupSort/Householder activation function $\phi$ is a continuous piecewise linear mapping by nature, and hence the solutions of (F.6) are well defined. Then we define the difference of two arbitrary trajectories of DEQ (F.6) as $\Delta z(t) := z(t) - z'(t) \in \mathbb{R}^d$. In general, we allow $z(t)$ and $z'(t)$ are generated by ODE (F.6) with different initial conditions. Next, we define $V(\Delta z(t)) := \Delta z(t)^\top \Pi \Delta z(t)$. Here, $V$ can be viewed as a weighted $\ell_2$ norm, i.e., $V(\Delta z(t)) = \langle \Delta z(t), \Delta z(t) \rangle_\Pi$ where $\langle \cdot, \cdot \rangle_\Pi$ denotes the $\Pi$-weighted inner product. For simplicity, we also denote $v_z(t) := \phi(Wz(t) + Ux + b)$. If (F.5) holds, then there exists a sufficiently small positive number $\epsilon$ such that the following non-strict matrix inequality also holds:

$$\begin{bmatrix} -2(1-\epsilon)\Pi & \Pi \\ \Pi & 0 \end{bmatrix} + \begin{bmatrix} W^\top(T-2S)W & W^\top(P+S) \\ (P+S)W & -T-2P \end{bmatrix} \preceq 0. \tag{F.7}$$

Based on the above matrix inequality, we immediately have

$$\begin{bmatrix} z(t) - z'(t) \\ v_z(t) - v_{z'}(t) \end{bmatrix}^\top \begin{bmatrix} -2(1-\epsilon)\Pi & \Pi \\ \Pi & 0 \end{bmatrix} \begin{bmatrix} z(t) - z'(t) \\ v_z(t) - v_{z'}(t) \end{bmatrix}$$
$$+ \begin{bmatrix} z(t) - z'(t) \\ v_z(t) - v_{z'}(t) \end{bmatrix}^\top \begin{bmatrix} W^\top(T-2S)W & W^\top(P+S) \\ (P+S)W & -T-2P \end{bmatrix} \begin{bmatrix} z(t) - z'(t) \\ v_z(t) - v_{z'}(t) \end{bmatrix} \leq 0. \tag{F.8}$$

It is easy to verify that the first term on the left side of (F.8) is equal to $\frac{\mathrm{d}}{\mathrm{d}t} V(\Delta z(t)) + 2\epsilon V(\Delta z(t))$. In addition, by Lemma 1/Lemma 2, the second term on the left side of (F.8) is guaranteed to be non-negative due to the properties of $\phi$. Therefore, we must have

$$\frac{\mathrm{d}}{\mathrm{d}t} V(\Delta z(t)) + 2\epsilon V(\Delta z(t)) \leq 0.$$

Now we can easily see that (F.6) is a contraction mapping with respect to the $\Pi$-weighted $\ell_2$ norm. The contraction rate is actually exponential and characterized by $\epsilon$. Since the flow map of our ODE is continuous and time-invariant, we can directly apply the contraction mapping theorem to prove the existence and uniqueness of the equilibrium point for (F.6). This immediately leads to the desired conclusion. □

Similar to Revay et al. (2020), it is possible to simplify the matrix inequality condition in Theorem F.2 via the KYP lemma and frequency-domain inequalities. Such developments are straightforward and omitted here. Given that the DEQ (F.4) is well-posed, we can combine the arguments in Revay et al. (2020) with Lemma 1 to obtain the following SDP conditions for analyzing the Lipschitz bounds of DEQ with GroupSort/MaxMin activations.

**Theorem F.3.** *Consider the DEQ (F.4) with $\phi$ being GroupSort with group size $n_g$. Suppose (F.4) is well-posed, and denote $N = \frac{d}{n_g}$. If there exist $\lambda \in \mathbb{R}_+^N$, $\gamma \in \mathbb{R}^N$, $\nu \in \mathbb{R}^N$ and $\tau \in \mathbb{R}^N$ such that*

$$\begin{bmatrix} W_o^\top W_o & 0 \\ 0 & -\rho I \end{bmatrix} + \begin{bmatrix} W^\top & I \\ U^\top & 0 \end{bmatrix} \begin{bmatrix} T-2S & P+S \\ P+S & -T-2P \end{bmatrix} \begin{bmatrix} W & U \\ I & 0 \end{bmatrix} \preceq 0. \tag{F.9}$$

*where $(T, S, P)$ are defined by equation (7) (equation (8)), then (F.4) is $\sqrt{\rho}$-Lipschitz from $x$ to $y$ in the $\ell_2 \to \ell_2$ sense.*

*Proof.* Given $x$ and $x'$, denote the corresponding hidden DEQ variables as $z$ and $z'$, respectively. Based on (F.9), we immediately have

$$\begin{bmatrix} z - z' \\ x - x' \end{bmatrix}^\top \begin{bmatrix} W_o^\top W_o & 0 \\ 0 & -\rho I \end{bmatrix} \begin{bmatrix} z - z' \\ x - x' \end{bmatrix}$$
$$+ \begin{bmatrix} z - z' \\ x - x' \end{bmatrix}^\top \begin{bmatrix} W^\top & I \\ U^\top & 0 \end{bmatrix} \begin{bmatrix} (T-2S) & (P+S) \\ (P+S) & -T-2P \end{bmatrix} \begin{bmatrix} W & U \\ I & 0 \end{bmatrix} \begin{bmatrix} z - z' \\ x - x' \end{bmatrix} \leq 0. \tag{F.10}$$

Based on (F.4) and Lemma 1/Lemma 2, the second term on the left side of (F.10) has to be non-negative. Therefore, the first term on the the left side of (F.10) has to be non-positive. Then we must have

$$\|W_o z + b_y - (W_o z' + b_y)\|_2 \leq \sqrt{\rho}\|x - x'\|_2,$$

which directly leads to the desired conclusion. $\qquad\square$

In Revay et al. (2020), SDPs that certify Lipschitz continuity have been used to derive parameterizations for DEQs with prescribed Lipschitz bounds. In a similar manner, our proposed SDP conditions can be used for network design. Further investigation is needed to address such research directions in the future.

### F.4 NEURAL ODE WITH GROUPSORT/HOUSEHOLDER ACTIVATIONS

In this section, we extend our approach to address another family of implicit learning models, namely neural ODEs (Chen et al., 2018). The neural ODE architecture can be viewed as a continuous-time analogue of explicit residual networks. Consider the following neural ODE:

$$\frac{\mathrm{d}}{\mathrm{d}t} z(t) = f_\theta(z(t), t) = G\phi\left(W_0 z(t) + W_1 t + b_0\right) + b_1, \quad z(0) = x \in \mathbb{R}^n, \qquad \text{(F.11)}$$

where $f_\theta : \mathbb{R}^n \times \mathbb{R} \to \mathbb{R}^n$ is a non-autonomous vector field parameterized as a neural network with trainable parameters $\theta = (G, W_0, W_1, b_0, b_1)$ and GroupSort/Householder activation $\phi$. The input $x$ to the neural ODE is used as the initial condition, and the output is just the solution of the resultant initial value problem at some final time $t_f > 0$.

Since $f_\theta$ is globally Lipschitz continuous in $z$ and $t$, we get a well-defined flow map $\Phi_0^t : \mathbb{R}^n \to \mathbb{R}^n$ which yields the following formula

$$\Phi_0^t(x) = z(t) = x + \int_0^t f_\theta(z(s), s)\mathrm{d}s.$$

For simplicity, we fix $t_f = 1$ and consider the neural ODE to be the map $\Phi_0^1$ that takes the input $z(0) = x$ to the final output $z(1) = \Phi_0^1(x)$. We can use Lemma 1/Lemma 2 to calculate Lipschitz bounds for the neural ODE flow map $\Phi_0^1$ as follows.

**Theorem F.4.** *Consider the neural ODE according to (F.11), where $\phi : \mathbb{R}^n \to \mathbb{R}^n$ is GroupSort (Householder) with group size $n_g$. Denote $N = \frac{n}{n_g}$. Suppose there exist $\rho > 0$, $\lambda \in \mathbb{R}_+^N$, $\gamma, \nu, \tau \in \mathbb{R}^N$ such that*

$$\begin{bmatrix} -\rho I & G \\ G^\top & 0 \end{bmatrix} + \begin{bmatrix} W_0^\top (T - 2S) W_0 & W_0^\top (P + S) \\ (P + S) W_0 & -T - 2P \end{bmatrix} \preceq 0, \qquad \text{(F.12)}$$

*where $(T, S, P)$ are given by equation (7) (equation (8)), then the neural ODE flow map $\Phi_0^1$ is $\exp(\rho/2)$-Lipschitz in the $\ell_2 \to \ell_2$ sense.*

*Proof.* Given two inputs $x$ and $x'$, we denote the corresponding trajectories as $z(t)$ and $z'(t)$, respectively. Then we define the difference of these two trajectories as $\Delta z(t) := z(t) - z'(t) \in \mathbb{R}^n$. In addition, we define $V(\Delta z(t)) := \|\Delta z(t)\|_2^2$ and $v_z(t) := \phi(W_0 z(t) + W_1 t + b_0)$. Based on condition (F.12), we immediately have

$$\begin{bmatrix} z(t) - z'(t) \\ v_z(t) - v_{z'}(t) \end{bmatrix}^\top \begin{bmatrix} -\rho I & G \\ G^\top & 0 \end{bmatrix} \begin{bmatrix} z(t) - z'(t) \\ v_z(t) - v_{z'}(t) \end{bmatrix}$$
$$+ \begin{bmatrix} z(t) - z'(t) \\ v_z(t) - v_{z'}(t) \end{bmatrix}^\top \begin{bmatrix} W_0^\top (T - 2S) W_0 & W_0^\top (P + S) \\ (P + S) W_0 & -T - 2P \end{bmatrix} \begin{bmatrix} z(t) - z'(t) \\ v_z(t) - v_{z'}(t) \end{bmatrix} \leq 0. \qquad \text{(F.13)}$$

It is easy to verify that the first term on the left side of (F.13) is equal to $\frac{\mathrm{d}}{\mathrm{d}t} V(\Delta z(t)) - \rho V(\Delta z(t))$. In addition, by Lemma 1 (Lemma 2), the second term on the left side of (F.13) is guaranteed to be non-negative for any $t$. Therefore, the following inequality holds for all $t \geq 0$:

$$\frac{\mathrm{d}}{\mathrm{d}t} V(\Delta z(t)) \leq \rho V(\Delta z(t)).$$

Finally, integrating $V$ up to time $t = 1$ yields

$$\|\Phi_0^1(x) - \Phi_0^1(x')\|_2 = \|z(1) - z'(1)\|_2 \leq \exp(\rho/2)\|x - x'\|_2,$$

which completes our proof. □

