# OpenReview forum: "Novel Quadratic Constraints for Extending LipSDP beyond Slope-Restricted Activations"
_ICLR.cc/2024/Conference — ICLR 2024 poster_

### Official Review · Reviewer_pJPd · 2023-10-29

**Soundness:** 3 good
**Presentation:** 3 good
**Contribution:** 2 fair
**Rating:** 6
**Confidence:** 3

**Summary:**

This paper extends the method of estimating the Lipschitz constant of a neural network using semidefinite programming (SDP) to the networks with non-slope-restricted activations functions such as GroupSort, Maxmin, and Householder. The SDP formulations are proposed for estimating $l_2$ and $l_\infty/l_1$ Lipschitz constants for various network architectures.

**Strengths:**

1. well-written, easy to follow even for a non-expert.
2. The extension of LipSDP to GroupSort, Maxmin, and Householder activations is new.

**Weaknesses:**

The main concern I have is that this paper seems to be an extension of two works Fazlyab'19 and Wang'22 to the case of having sum-preserving activations like GroupSort, Maxmin, and Householder, which seems incremental.

**Questions:**

1. How frequently are GroupSort, Maxmin, and Householder being used in practice? If they are not so popular, why we are studying LipSDP for them?
2. This is merely a comment. The results would be more interesting and valuable if using GroupSort, Maxmin, and Householder activations have some implicit bias towards having an NN with a smaller Lipschitz constant.

---

> ### Author Response · Authors · 2023-11-17
> **Response to Reviewer pJPd**
>
> Thank you for your evaluation and your feedback. In the following, we address your concerns and your question. Plase do let us know if you have follow-up questions, additional concerns or suggestions.
>
> > The main concern I have is that this paper seems to be an extension of two works Fazlyab'19 and Wang'22 to the case of having sum-preserving activations like GroupSort, Maxmin, and Householder, which seems incremental.
>
> We think that our contribution in deriving new quadratic constraints for GroupSort, MaxMin, and Householder are novel and significant, complementing the existing works such as Fazlyab'19 and Wang'22. We provide a detailed response in our general response (Part 1). Let us restate some of our key points here. We also included Appendix B to discuss the issue in our revised paper. The underlying idea in the quadratic constraint approach is to abstract "challenging", e.g., nonlinear, uncertain or time-varying elements using quadratic constraints. This abstraction allows us to formulate an SDP that can be solved effectively. Since the introduction of the quadratic constraint framework in the 1960s, many papers appeared that are only devoted to deriving new quadratic constraints. For example, Kao'07 derives quadratic constraints for a delay-difference operator to describe varying time delays, Pfifer'15 uses a geometric interpretation to derive new quadratic constraints for delayed nonlinear and parameter-varying systems, and Carrasco'16 summarizes the development of Zames-Falb multipliers based on frequency domain arguments for slope-restricted nonlinearites. In the context of NNs, so far only quadratic constraints for slope-restricted nonlinearities were utilized. By formulating quadratic constraints for GroupSort/Householder activations, we are the first ones to study multivariate activations, to formulate quadratic constraints for NNs that go beyond slope-restriction or Lipschitz continuity and in general the first to present quadratic constraints for sum-preserving elements. Finding novel quadratic constraints is by no means trivial. It requires to study and understand the characteristics of the underlying nonlinearity and then to capture the identified property using a quadratic form. There is no recipe for such a derivation and finding a quadratic constraint does not guarantee that it improves the analysis over other methods. The tighter the description of the nonlinearity by quadratic constraints, the better the analysis in terms of conservatism. The idea we used to derive these quadratic constraints is creative in the sense that it is very different from all the existing quadratic constraint derivations, hence offering unique new insights and complementing the large body of existing arguments in deriving quadratic constraints.  To understand the technical details of our derivation, we refer to the proofs in Appendices C.2 and C.3.
>
> > How frequently are GroupSort, Maxmin, and Householder being used in practice?
>
> Please see our general response (Part 2). GroupSort, MaxMin, and Householder activations are gradient norm preserving and have been of particular interest when fitting Lipschitz functions and for the design of certifiably robust neural networks (NNs). There are many works (e.g. Leino'21, Singla'21, Trockman'21, Huang'21, Singla'22, Prach'22, Hu'23) that use MaxMin activations and provide state-of-the-art certified robust accuracy (please see our general response, Part 2 for a detailed list of the references). To compute the certified robust accuracy, these works rely on information of Lipschitz bounds and prediction margins. Hence LipSDP for MaxMin networks is of great interests. Another interesting application of MaxMin NNs is in learning-based control (Yion'23), where the information of the Lipschitz constant of a neural network in the loop can be useful to verify safety and robustness. Please note that we included a section on applications of MaxMin NNs as Appendix A.
>
> > The results would be more interesting and valuable if using GroupSort, Maxmin, and Householder activations have some implicit bias towards having an NN with a smaller Lipschitz constant.
>
> The implicit bias towards smaller Lipschitz constant of neural networks is typically governed by specific neural network structures (e.g. Singla'21, Trockman'21, Prach'22) or regularization terms in the training objective (e.g. Leino'21,Huang'21, Hu'23), but not directly through the choice of the activation function. The popularity of the MaxMin activation (or GroupSort and Householder in general) for certifiably robust networks is mainly due to their gradient norm preserving (GNP) properties, since the GNP properties can help the training of certifiably robust neural networks become more stable.

---

> > ### Comment · Reviewer_pJPd · 2023-12-01
> > **Thanks for the response**
> >
> > I think the author has addressed the importance of estimating the Lipschtiz constant of networks with Maxmin, groupsort activations. Therefore I have increased my score.

---

### Official Review · Reviewer_8tsi · 2023-10-31

**Soundness:** 3 good
**Presentation:** 3 good
**Contribution:** 3 good
**Rating:** 6
**Confidence:** 2

**Summary:**

This paper presents new semi-definite programs for computing upper-bounds on
the Lipschitz constants of deep neural networks with gradient-preserving
activations.  The authors derive new quadratic constraints which extend the
state-of-the-art LipSDP framework to Lipschitz estimation with the GroupSort,
MaxMin, and Householder activation functions; these activations were previously
not covered by LipSDP since they do not satisfy the slope-restricted property.
The authors then extend their approach to compute Lipschitz constants in the
$\ell_\infty$ norm and show how to apply their results to neural networks with
residual connections. Experiments confirm the empirical performance of the
proposed SDPs.

**Strengths:**

This is an interesting submission which extends the existing LipSDP framework
for estimating Lipschitz constants of neural networks to new activation
functions. The authors use the quadratic constraint approach from control
theory to obtain polynomial-time algorithms for the GroupSort and Householder
activations (these generalize the MaxMin activation). As only naive estimation
approaches previously existed for these activations, this contribution is fairly
strong and represents the major strength of this paper.

Other notable strengths are the following:

- The proposed SDPs yield upper-bounds for small networks which are close to
    those obtained by brute-force search over the activation space, particularly
    for the $\ell_2$-norm. Moreover, the bounds are much tighter than those
    obtained using operator norms.

- The methodology is presented clearly and the manuscript is polished.

**Weaknesses:**

The major limitation of this work is the restriction to Householder and
GroupSort activations. The utility of extending the LipSDP framework to these
activation depends directly on the how interesting the problem of Lipschitz
estimation is for neural networks using these architectures. While the authors
state that such activations are becoming popular for the design of Lipschitz
neural networks, no concrete examples are provided. I am also concerned about
the following:

- The basic idea of LipSDP was developed by Fazlyab et al. (2019) while the
    extension to estimation in the $\ell_\infty$ norm is from Wang et al. (2022).
    The main theoretical contribution of this work is to develop new quadratic
    constraints which fit into those frameworks, rather than build significantly
    on top of them. Thus, the paper may be somewhat incremental in nature.

- The authors do not provide the computation time for the naive baseline method
    for approximating Lipschitz constants based on operator norms (MP),
    so it is not clear what the trade-off between computation and accuracy is
    for the proposed method.

I am hesitant to recommend this submission for acceptance without additional
evidence that the Householder and GroupSort activations are of practical
interest for Lipschitz estimation (see "Questions").
Moreover, this paper is outside of my research area so it is difficult for me
to judge its theoretical novelty; I did not check the proofs for correctness
for the same reason. Given this, and the smaller issues raised above, I am
 on the fence regarding this submission.

**Questions:**

As noted above, I am not a expert on Lipschitz constant estimation for neural
networks nor have I made use of algorithms from this area. Given this, can the
authors please provide additional details on why the GroupSort and Householder
activations are of particular interest for Lipschitz constant estimation? Since
these activations are the exclusive focus of the paper, I feel there must be an
immediate desire from the community to solve this problem in practice for
the paper to have a significant impact.

I would also appreciate it if the authors could provide running times for the
naive estimation strategies in Table 1; this well help contextualize the cost
of LipSDP-NSR and clarify the trade-off between accuracy and compute time.

Finally, perhaps the authors can comment on the difficulty of deriving
the quadratic constraints for the SDPs. This will help me understand the novelty
of the theoretical contributions.

---

> ### Author Response · Authors · 2023-11-17
> **Response to Reviewer 8tsi**
>
> Thank you for your thorough evaluation of our paper. We would like to address your comments as below. We are also available to address any potential follow-up questions/suggestions.
>
> > Additional details on why the GroupSort and Householder activations are of particular interest for Lipschitz constant estimation?
>
>  Please see our general response (Part 2). GroupSort and Householder activations are gradient norm preserving and have been of particular interest when fitting Lipschitz functions and for the design of certifiably robust neural networks (NNs). There are many works published in top machine learning conferences (e.g. Leino'21, Singla'21, Trockman'21, Huang'21, Singla'22, Prach'22, Hu'23) that use MaxMin activations and provide state-of-the-art certified robust accuracy (we provide details in our general response, Part 2). To compute the certified robust accuracy, these works rely on information of Lipschitz bounds and prediction margins of the trained networks. Hence LipSDP for MaxMin networks is of great interests. Another interesting application of MaxMin NNs is in learning-based control (Yion'23), where the information of the Lipschitz constant of a neural network in the loop can be useful to verify safety and robustness. Please note that we included a section on applications of MaxMin NNs as Appendix A.
>
> > I would also appreciate it if the authors could provide running times for the naive estimation strategies.
>
> We are happy to provide this additional information. We recalculated the computation times for all our naive bounds. The computation times of the matrix product bound are for all our networks below 10s, so they are basically instantaneous. This means there definitely is a price for the tighter bounds using our method. The computation times for the lower bounds from sampling based on 200k samples range between 12 and 23 s. The FGL bound takes 4.54s for the 2FC-16 NN and 81.92s for the 2FC-32 NN. For larger NNs it however becomes intractable (we stopped our code for 2FC-64 after 4 hours).
>
> Find the running times of all naive l2 bounds for our networks in Table 1 in the table below:
> | Architecture  | Sample   | FGL | MP |
> | -- |:--:| --:|:--:|
> | 2FC-16        | 12.02 | 4.54 | 6.16 |
> | 2FC-32        | 13.08 | 81.92| 5.33 |
> | 2FC-64        | 12.84 |      | 3.95 |
> | 2FC-128       | 12.61 |      | 3.6  |
> | 5FC-32        | 13.18 |      | 5.43 |
> | 5FC-64        | 14.83 |      | 5.68 |
> | 8FC-32        | 14.14 |      | 5.57 |
> | 8FC-64        | 14.27 |      | 5.6  |
> | 8FC-128       | 14.15 |      | 9.85 |
> | 8FC-256       | 13.74 |      | 9.3  |
> | 18FC-32       | 23.28 |      | 7.64 |
> | 18FC-64       | 21.09 |      | 9.22 |
> | 18FC-128      | 19.48 |      | 4.9  |
>
> > Finally, perhaps the authors can comment on the difficulty of deriving the quadratic constraints for the SDPs.
>
> Please see our general response (Part 1) for a detailed response. We are also happy to restate some of our key points here. We also included Appendix B to discuss this issue in our revised paper. The underlying idea in the quadratic constraint approach is to abstract "challenging", e.g., nonlinear, uncertain or time-varying elements using quadratic constraints. This abstraction allows us to formulate an SDP that can be solved effectively. Since the introduction of the quadratic constraint framework in the 1960s, many papers appeared that are only devoted to deriving new quadratic constraints. For example, Kao'07 derives quadratic constraints for a delay-difference operator to describe varying time delays, Pfifer'15 uses a geometric interpretation to derive new quadratic constraints for delayed nonlinear and parameter-varying systems, and Carrasco'16 summarizes the development of Zames-Falb multipliers based on frequency domain arguments for slope-restricted nonlinearites. In the context of NNs, so far only quadratic constraints for slope-restricted nonlinearities were utilized. By formulating quadratic constraints for GroupSort/Householder activations, we are the first ones to study multivariate activations, to formulate quadratic constraints for NNs that go beyond slope-restriction or Lipschitz continuity and in general the first to present quadratic constraints for sum-preserving elements. Finding novel quadratic constraints is by no means trivial. It requires to study and understand the characteristics of the underlying nonlinearity and then to capture the identified property using a quadratic form. There is no recipe for such a derivation and finding a quadratic constraint does not guarantee that it improves your analysis over other methods. The tighter your description of the nonlinearity by quadratic constraints, the better the analysis in terms of conservatism. The idea we used to derive new quadratic constraints is creative in the sense that it is very different from all the existing quadratic constraint derivations, hence offering unique new insights on how to use properties such as sum-preservation.

---

> ### Comment · Reviewer_8tsi · 2023-11-20
>
> Thanks for responding to my concerns and for providing the timings of the naive baseline methods. I also appreciate the inclusion of LipSDP-RR along with runtimes. I think these results provide a more nuanced view of LipSDP-NSR and fit well with the paper's story, which was motivated by better bounds in the L2 norm. I am not particularly concerned that LipSDP-RR sometimes has better performance for the L-infinity norm since computing the constant in the L-infinity norm is most "value added" in my opinion.
>
> It is surprising to me that FGL is faster than LipSDP-NSR for small networks despite being an exponential time algorithm. Of course, this time complexity catches up for larger models as you mention. I suppose the constant-factor overhead for the SDP solver dominates when running on small architectures.
>
> The matrix-product approach (MP), while naive, is very fast and provides acceptable-looking bounds when the number of layers is eight or less. I think it's worth including the runtimes you provided here in the paper (appendix is fine) and including a comment on the trade-off between efficiency and tightness for MP. In many cases it may be worth obtaining a loose bound quickly, which MP does well.
>
> Overall, I am convinced that the paper has practical use and that the quadratic constraints are of theoretical interest, so I will increase my score to 6. However, I do feel this paper is closer to 7 (although this doesn't exist for ICLR) and lean towards accepting the submission.

---

> > ### Author Response · Authors · 2023-11-21
> > **Response to Reviewer 8tsi**
> >
> > Thank you for your response and positive reevaluation of our work and for your suggestion to address the trade-off between efficiency and tightness. We agree that LipSDP-RR fits well into the story and therefore, we gladly included it in our revised paper. We have also included the computation times of our baseline methods as Appendix D.3 in the revised paper and discussed the trade-off between computational efficiency and accuracy of the bounds in the same section as suggested. In the last paragraph of Appendix D.3, we commented that the computation of the matrix product bound (MP), while naive, is quite fast, taking less than 10 s for all our models. We also mentioned that MP provides reasonably accurate bounds when the number of layers is small, but becomes very loose for deep neural networks.

---

### Official Review · Reviewer_ykPt · 2023-11-01

**Soundness:** 4 excellent
**Presentation:** 3 good
**Contribution:** 2 fair
**Rating:** 6
**Confidence:** 3

**Summary:**

This paper considered the problem of estimating the Lipschitz constant of neural networks with different kinds of activations that are not slope-restricted. Particularly, the authors investigated multi-layer (residual) networks applied with the GroupSort and Householder activations. The paper followed the idea of thge LipSDP formulation of the Lipschitz parameter estimation problem, and the main contribution is that the authors devised a new quadratic constrained that can deal with GroupSort and Householder which are not slope-restricted. In addition, the authors conducted empirical experiments which showed that the new formulation with quadratic cosntraints outperforms traditional matrix-product algorithms in terms of the accuracy of the estimated Lipschitz parameter.

**Strengths:**

The paper is written clearly and the authors provided useful intuiation.

The new quadratic constraints enabled us to estimate the Lipschitz constants of neural networks with GroupSort or Householder activations with higher accuracy compared to traditional algorithms.

In addition, the authors considered both $\ell_2$ and $\ell_\infty\to\ell_1$ lipschitz constants.

**Weaknesses:**

The result seems to be weak since the quadratic constraints only applied to 2 specific activations. It would be more interesting if it can be applied to a class of activaitons.

**Questions:**

How does the new formaulation compare to the original LipSDP for MaxMin networks in terms of computational efficiency/runtime?

---

> ### Author Response · Authors · 2023-11-17
> **Response to Reviewer ykPt**
>
> We appreciate your comments and evaluation of our work. We will address your concerns in the following and are happy to answer any follow-up questions.
>
> >The result seems to be weak since the quadratic constraints only applied to 2 specific activations. It would be more interesting if it can be applied to a class of activaitons.
>
> We would consider GroupSort and HouseHolder as two classes of activations. Both generalize MaxMin which is widely used for robust networks due to the ease of implementation. Our constructions of quadratic constraints are also general in the sense that our GroupSort quadratic constraint also holds for the class of all sum-preserving activations. We now clearly state this in introduction of the revised version of the paper.
>
> >How does the new formaulation compare to the original LipSDP for MaxMin networks in terms of computational efficiency/runtime?
>
> Thank you for this question. To investigate this question, we used the description of the MaxMin acitvation using a residual ReLU network (4), as shown in our motivating example in Section 3. For this equivalent formulation we then implemented an SDP for the residual ReLU equivalent form of MaxMin (the multi-layer variants of (5) under either $\ell_2$ or $\ell_\infty$ setting) using quadratic constraints for slope-restricted activations to compute Lipschitz bounds. We denote the SDP on the residual ReLU formulation by LipSDP-RR. We added our results in the paper and for convenience also state the new results in the table below, where we compare the bounds and the computation times. We observe that our LipSDP-NSR provides better bounds in most cases, with two exceptions for 2-layer networks under the $\ell_\infty$ setting, i.e. $\ell_\infty$ Lipschitz bounds for 2FC-64 and 2FC-128. For 5- and 8-layer networks, our LipSDP-NSR is always better than LipSDP-RR, and there even is a significant gap.  Notice that LipSDP-NSR outperforms LipSDP-RR in all cases under the $\ell_2$ setting. This is consistent with the intuition provided by our motivating example in Section 3 of our paper. Specifically, our motivating example in Section 3 implies that  the quadratic constraints derived from slope-restricted properties of ReLU have severe fundamental limitations in addressing the $\ell_2$ case of MaxMin Lipschitz analysis, i.e. it cannot recover the basic 1-Lipschitz property of the MaxMin activation under the $\ell_2$ setting. Our new quadratic constraints derived for MaxMin fix that, and our numerical results support our claim. In the $\ell_\infty$ setting, the output is always a scalar and the 1-Lipschitzness of MaxMin activation is not explicitly needed for SDP-based analysis. Hence our motivating example does not indicate much for the $\ell_\infty$ setting. However, our LipSDP-NSR still outperforms LipSDP-RR in most cases under the $\ell_\infty$ setting. Finally, we want to comment on the computation efficiency. We want to point out that advantages in computational efficiency of the MaxMin quadratic constraint formulation become apparent only for larger networks. It is interesting to notice that for small neural networks, LipSDP-RR can sometimes be faster, and this can be explained by the different implementations of the two problems. The computation time depends on the number of decision variables and on the conditioning of the problem. While the number of decision variables are the same in both problems, the multiple constraints for LipSDP-NSR are much smaller than the one constraint in LipSDP-RR. For networks larger than 8FC-64 we even run into memory problems when solving the SDP for the residual ReLU network formulation.
>
> | Architecture  | L-l2 (Time) NSR | L-l2 (Time) RR  | L-linfty (Time) NSR | L-linfty (Time) RR |
> | ------------- |:------------:| ------------:|:------------:| -----------:|
> | 2FC-16        | 22.59 (54)   | 23.92 (18)   | 203 (94)     | 207.3 (34)  |
> | 2FC-32        | 38.55 (86)   | 40.28 (24)   | 368.7 (128)  | 404.8 (44)  |
> | 2FC-64        | 73.77 (160)  | 74.31 (23)   | 885.9 (147)  | 870.8 (38)  |
> | 2FC-128       | 70.86 (182)  | 72.18 (55)   | 1058.8 (236) | 1033.1 (77) |
> | 5FC-32        | 224.6 (53)   | 388.0 (69)   | 2969.5 (96)  | 4283.1 (137)|
> | 5FC-64        | 386.8 (111)  | 623.4 (228)  | 6379.8 (190) | 8321.3 (416)|
> | 8FC-32        | 401.0 (58)   | 1078.5 (178) | 4303.9 (120) | 8131.1 (374)|
> | 8FC-64        | 722.3 (116)  | 1683.7 (1426)| 9632.3 (188) | 1.886E+04 (2495)|

---

> > ### Comment · Reviewer_ykPt · 2023-11-21
> >
> > I thank the authors for their feedback and their new experiments that showcase the efficiency of the LipSDP-NSR. I think the authors responded to my questions properly and detailedly. I would like to keep my scores unchanged.

---

### Author Response · Authors · 2023-11-17
**General Response, Part 1**

We thank all reviewers for their valuable feedback. Let us start with a general response to address two main concerns that were raised by the reviewers and clarify some doubts/questions. Firstly, the reviewers were unsure about the **difficulty and significance of deriving new quadratic constraints for GroupSort/Householder**. Secondly, the reviewers also asked about **where Lipschitz constant estimation for MaxMin neural networks is needed**. We address these two points as follows. (We have added some new references to our revised paper to support our arguments. Please see the revised paper's list for details. We are available to answer any more questions and address any more concerns or suggestions the reviewers may have throughout the rebuttal.)

# Significance and novelty of our contribution

As recognized by the reviewers, our main technical contribution is the development of new quadratic constraints for GroupSort/Householder activations. Developing quadratic constraints is a highly non-trivial task, and there are many useful papers which completely focus on developing quadratic constraints. The quadratic constraint framework stems from the control community and was developed starting in the 1960s. The idea is that for the purposes of robustness analysis, one can abstract troublesome elements (e.g., nonlinearities, uncertainties, or time delays) using quadratic constraints on their input/output behaviors. This leads to a **modular**  analysis framework that can include various troublesome elements as long as their input/output behaviors can be captured by meaningful quadratic constraints. Over the years, the quadratic constraint approach has become powerful due to significant contributions along the following two lines of research:
- The first line of research focuses on answering this question: given quadratic constraints, how can we formulate SDP conditions to address a specific type of robustness analysis? The original seminal work in  Megretski'97 is of this nature. Similarly, Fazlyab'19 and Wang'22 are also of this nature.

- The second research line studies how to develop quadratic constraints for various types of troublesome elements such as nonlinearities, uncertainties, and time delays. Since there is no general recipe for developing quadratic constraints, many novel papers actually completely focus on how to derive quadratic constraints. For example, Kao'07 introduces quadratic constraints for a so-called "delay-difference" operator to describe time-varying delays, Pfifer'15 uses a geometric interpretation to derive quadratic constraints for delayed nonlinear and parameter-varying systems, and Carrasco'16 summarizes the development of Zames-Falb multipliers for slope-restricted nonlinearites based on frequency domain arguments.



The above papers along the two lines of research are both novel, and complement each other. **Importantly, our paper is the first that successfully derives quadratic constraints for GroupSort and Householder activations, and further, we are the first to exploit the sum-preserving properties for deriving quadratic constraints on nonlinearity beyond the slope-restricted properties.** The difficulty of deriving quadratic constraints lies in identifying properties and characteristics of the troublesome element, in our case a multivariate nonlinearity GroupSort or Householder, that can (i) be formulated in a quadratic form and (ii) whose quadratic constraint formulation is descriptive enough to lead to improvements in the analysis. Please see C.2 and C.3 (in our revised paper) for the technical details of our derivation. All previous papers on LipSDP borrow existing quadratic constraints for slope-restricted nonlinearities from the control literature. There are no such quadratic constraints available for  GroupSort and Householder. The idea we used to derive these quadratic constraints is creative in the sense that it is very different from all the existing quadratic constraint derivations in the large body of control literature, hence offering unique new insights and complementing the large body of existing arguments in deriving quadratic constraints. Therefore, we think that it is fair to claim that our main contribution of developing quadratic constraints for GroupSort/HouseHolder activations is technically novel.

---

> ### Author Response · Authors · 2023-11-17
> **General Response, Part 2**
>
> # Practical Use of MaxMin and Relevance of Lipschitz Estimation of MaxMin Networks
>
> In the deep learning field, MaxMin activations have been widely used to address certified robustness of neural networks. The following papers are all published in top machine learning venues and use MaxMin to improve certified robust accuracy of deep neural networks.
>
>
>
> [Leino'21] Klas Leino, Zifan Wang, and Matt Fredrikson. Globally-robust neural networks. ICML
>
> [Singla'21]. Sahil Singla and Soheil Feizi. Skew orthogonal convolutions. ICML
>
>
> [Trockman'21] Asher Trockman and J Zico Kolter. Orthogonalizing convolutional layers with the Cayley transform. ICLR
>
> [Huang'21] Yujia Huang, Huan Zhang, Yuanyuan Shi, J Zico Kolter, and Anima Anandkumar. Training certifiably robust neural networks with efficient local Lipschitz bounds. NeurIPS
>
> [Singla'22] Sahil Singla, Surbhi Singla, and Soheil Feizi. Improved deterministic l2 robustness on CIFAR-10 and CIFAR-100. ICLR
>
> [Prach'22] Bernd Prach and Christoph H Lampert. Almost-orthogonal layers for efficient general-purpose lipschitz networks. ECCV
>
> [Hu'23] Kai Hu, Andy Zou, Zifan Wang, Klas Leino, and Matt Fredrikson. Unlocking deterministic robustness certification on imagenet. NeurIPS
>
> The above papers provide state-of-the-art (deterministic) certified robust accuracy for neural networks, and all these works use MaxMin as activation functions. The state-of-the-art certified robust accuracy for Imagenet is actually achieved using MaxMin activations (Hu'23), and Huang'21 for example states that MaxMin can achieve significantly better results on certified robust accuracy than its ReLU counterpart. **To compute the certified robust accuracy, one needs to use the information of the Lipschitz constant and the prediction margin of the trained networks.** Therefore, studying LipSDP for improving the Lipschitz constant estimation of MaxMin networks is of great interest to the deep learning community.
>
> In addition, LipSDP for MaxMin networks can also be useful for learning-based control with neural network controllers. Very recently, researchers have started to use MaxMin activations for Lipschitz neural network controllers in this field (Yion'23). LipSDP for MaxMin can potentially reduce the conservatism in the stability and robustness analysis of such MaxMin network controllers.

---

### Meta-Review · Area_Chair_WD2H · 2023-12-18

**Metareview:**

This paper investigates the problem of providing accurate Lipschitz bounds for neural networks. It extends the LipSDP approach to develop new quadratic constraints for GroupSort, MaxMin, and Householder activations in neural networks. Moreover, numerical results are provided to validate the proposed approach. The reviewers were generally positive about the paper.

**Justification For Why Not Higher Score:**

The proposed approach is still computationally costly, which limits its applicability.

**Justification For Why Not Lower Score:**

The proposed approach can handle a broader class of activation functions than the results in the literature, which is of theoretical interest.

---

### Decision · Program_Chairs · 2024-01-16

Accept (poster)